# Cu$_2$Se-based thermoelectric cellular architectures for efficient and durable power generation

Seungjun Choo[1,5], Faizan Ejaz[2,5], Hyejin Ju[1], Fredrick Kim[1], Jungsoo Lee[1], Seong Eun Yang[1], Gyeonghun Kim[3], Hangeul Kim[1], Seungki Jo[1], Seongheon Baek[1], Soyoung Cho[1], Keonkuk Kim[1], Ju-Young Kim[1], Sangjoon Ahn [3], Han Gi Chae [1✉], Beomjin Kwon [2✉] & Jae Sung Son [1,4✉]

Thermoelectric power generation offers a promising way to recover waste heat. The geometrical design of thermoelectric legs in modules is important to ensure sustainable power generation but cannot be easily achieved by traditional fabrication processes. Herein, we propose the design of cellular thermoelectric architectures for efficient and durable power generation, realized by the extrusion-based 3D printing process of Cu$_2$Se thermoelectric materials. We design the optimum aspect ratio of a cuboid thermoelectric leg to maximize the power output and extend this design to the mechanically stiff cellular architectures of hollow hexagonal column- and honeycomb-based thermoelectric legs. Moreover, we develop organic binder-free Cu$_2$Se-based 3D-printing inks with desirable viscoelasticity, tailored with an additive of inorganic Se$_8^{2-}$ polyanion, fabricating the designed topologies. The computational simulation and experimental measurement demonstrate the superior power output and mechanical stiffness of the proposed cellular thermoelectric architectures to other designs, unveiling the importance of topological designs of thermoelectric legs toward higher power and longer durability.

[1] Department of Materials Science and Engineering, Ulsan National Institute of Science and Technology (UNIST), Ulsan, Republic of Korea. [2] School for Engineering of Matter, Transport and Energy, Arizona State University, Tempe, AZ, USA. [3] Department of Nuclear Engineering, Ulsan National Institute of Science and Technology, Ulsan, Republic of Korea. [4] Center for Future Semiconductor Technology (FUST), Ulsan National Institute of Science and Technology (UNIST), Ulsan, Republic of Korea. [5] These authors contributed equally: Seungjun Choo, Faizan Ejaz. ✉email: hgchae@unist.ac.kr; kwon@asu.edu; jsson@unist.ac.kr

With the rapid increase in global energy consumption, the world is facing serious energy and environmental crises as a result of fossil fuel depletion and environmental pollution, critically requiring sustainable and renewable energy sources. Among the various types of renewable energy sources, thermal energy is omnipresent in natural and artificial environments, but more than 60% is dissipated. Thermoelectric (TE) power generation has been regarded as a reliable and durable way to recover dissipated waste heat, as it enables the direct conversion of heat to electricity without any environmental pollution[1–5]. Thus far, to achieve an efficient TE energy conversion, considerable efforts have been devoted toward the development of efficient TE materials, including traditional $Bi_2Te_3$-[6–9], PbTe-[10], and SiGe-based[11] alloys and emerging materials of liquid-like materials[12–15], skutterudites[16], half-Heuslers[17], SnSe[18,19], MgSb-based materials[20], etc.[21–23]. Among them, liquid-like TE materials within the "phonon-glass electron-crystal" concept have attracted tremendous attention due to their high efficiencies arising from ultralow thermal conductivities at high temperatures. Such unusual properties originate from their structural characteristics in which one immobile ion forms a rigid sublattice framework for the free transport of electrons, whereas other mobile ions are comprised of liquid-like sublattice to interrupt the thermal transports. Moreover, $Cu_2Se$- or $Cu_{1.97}S$-based compounds representing this class of materials have nontoxic and earth-abundant elements, which significantly reduce the cost by at least one order of magnitude compared with state-of-the-art TE materials[12–14].

In addition to the development of materials, the module structural design, especially the geometrical design of TE legs, is another essential factor in the efficiencies, durability, and cost of thermoelectric generators (TEGs). Numerous numerical simulations have demonstrated the remarkable effect of the three-dimensional (3D) geometries of TE legs on the energy conversion efficiency in a module and proposed various new designs of structures of TE legs that aim to enhance the efficiency[24–28]. However, most studies have focused only on efficiency maximization, neglecting the mechanical durability and cost, despite their actual importance. Moreover, experimental research on the geometrical design of TE legs still remains in the early stages in that the basic geometrical parameters, such as the aspect ratio or cross-sectional area ratio of stereotypical cuboid-shaped TE legs, have been investigated[29,30]. This geometrical limitation of TE legs may originate from the conventional process of the top-down dicing of TE ingots to produce TE legs, where there is no choice of shapes in a TE leg other than a cuboid.

Nature-inspired architectured cellular materials are an emerging class of materials with high stiffness, controllable heat dissipation and transfer, and lightweightness, which allow potential applications in highly stiff panels, energy absorbers, heat exchangers, vibration damping, and catalysts[31–33]. Cellular materials generally consist of an interconnected network of solid structures formed by cell walls separated by periodic or stochastic pores. The shape and layout of pores are critical to the mechanical properties of the cellular materials, since they determine the effective density and deformation modes. For example, in stochastic open-cell structures, structural Young's modulus decreases with density as an empirical power law with an exponent of two to three. However, in periodic closed-cellular structures such as honeycomb architecture, both the mechanical strength and stiffness linearly decrease with the relative density[34–36]. Thus, the closed-cellular structures tend to exhibit greater mechanical performance than open-cellular structures when their densities are reduced in a similar amount[37]. Honeycombs represent the two-dimensional (2D) closed-cellular architectures used almost exclusively today due to their relatively simple structures and excellent mechanical properties, such as high in-plane compression and out-of-plane shear properties[38]. When compared with triangular and hexagonal truss-structured materials with similar densities,

optimally designed honeycombs exhibited multi-fold enhancement in compressive strength. The honeycomb architecture resisted buckling upon compression unlike the truss-based counterparts[39]. Recently, the specific stiffnesses of ceramic hexagonal and triangular honeycombs were reported as $>10^7$ Pa kg$^{-1}$ m$^{-3}$ that surpass other micro- and nanoscale lattices of similar relative densities[40]. The 3D printing process has been recognized as an advanced technology for directly producing such 3D cellular architectures with great geometrical complexity in a cost-effective manner[41,42]. However, the complex 3D geometries of TE materials and modules have never been realized so far because the full functionality of 3D printing technology has yet not been applied to TE technology, though many approaches based on stereolithography, extrusion-based printing, and selective laser sintering have been reported to produce TE materials[43–49]. Moreover, in most reports, 3D-printable materials have still been limited to $Bi_2Te_3$-based materials, requiring the expansion of available materials operatable at high temperatures for the widespread applications of this technology[44–48].

Here, we designed the cellular honeycomb topology of $Cu_2Se$ TE legs by the 3D finite element models (FEMs) for higher power-generating performances and stronger mechanical stiffness than a typical cuboid. To fabricate the designed topology, we developed the extrusion-based 3D printing process of $Cu_2Se$ TE materials. The extrusion-based 3D printing process has been extensively studied for the production of inorganic 3D objects using concentrated colloid inks. In this process, colloid inks containing semiconductor metal chalcogenides must have the desired rheological properties to ensure 3D printability. To this end, we designed the $Cu_2Se$ colloid ink to have particles with surface charges, achieved by the addition of $Se_8^{2-}$ polyanion, showing significantly improved printability. Moreover, $Se_8^{2-}$ polyanion acts as a sintering promoting aid, which leads to the liquid-phase sintering of $Cu_2Se$ particles to shape and scale robust $Cu_2Se$ materials with the competitive $ZT$ value of 1.2 at 1000 K by the 3D printing process. The fabrication and characterization of the power-generating modules chipped with a cuboid, a hollow hexagonal column, and a honeycomb showed the highest power performance of a honeycomb-architectured TE leg, demonstrating the feasibility of our nature-inspired design toward the fabrication of efficient and durable TEGs.

## Results

**Design of $Cu_2Se$ TE architectures.** To design the topology of $Cu_2Se$ TE legs toward higher power generating performance as well as mechanical durability, we developed a 3D FEM to calculate $P$, $\Delta T$, and electrical resistance of a TEG (Fig. 1a–d, and Supplementary Fig. 1). Generally, the major geometrical control parameter of a TE leg is an aspect ratio in a cuboid since the aspect ratio dependence on the output power has the trade-off relation between a module resistance and a temperature difference ($\Delta T$) across a TE leg[29,30,50]. Accordingly, we define the aspect ratio as the ratio of the leg length ($l$) to cross-sectional area ($A$) (Fig. 1a and Supplementary Fig. 1a). In this computation, we used the TE properties of the 3D-printed $Cu_2Se$ TE materials. The detailed TE properties are discussed in the following sections. Based on the 3D FEM, a $Cu_2Se$ cuboid TE leg is expected to produce the maximum $P$ near $l/A$ of 1.5 at a various hot-side temperatures ranging from 700 to 850 K (Fig. 1b). The optimum $l/A$ results from the characteristic dependence of $\Delta T$ and resistance on $l/A$. If $l/A$ varies from 0.5 to 10, $\Delta T$ increases in a log-linear manner while the electrical resistance increases linearly with $l/A$ (Supplementary Fig. 2a, 2b).

In addition to the power output, the topology control of a TE leg can be beneficial to enhance the mechanical strength, which is critical for the practical use of TEGs. In a real-world application, TE legs in a TEG are forced to experience various mechanical

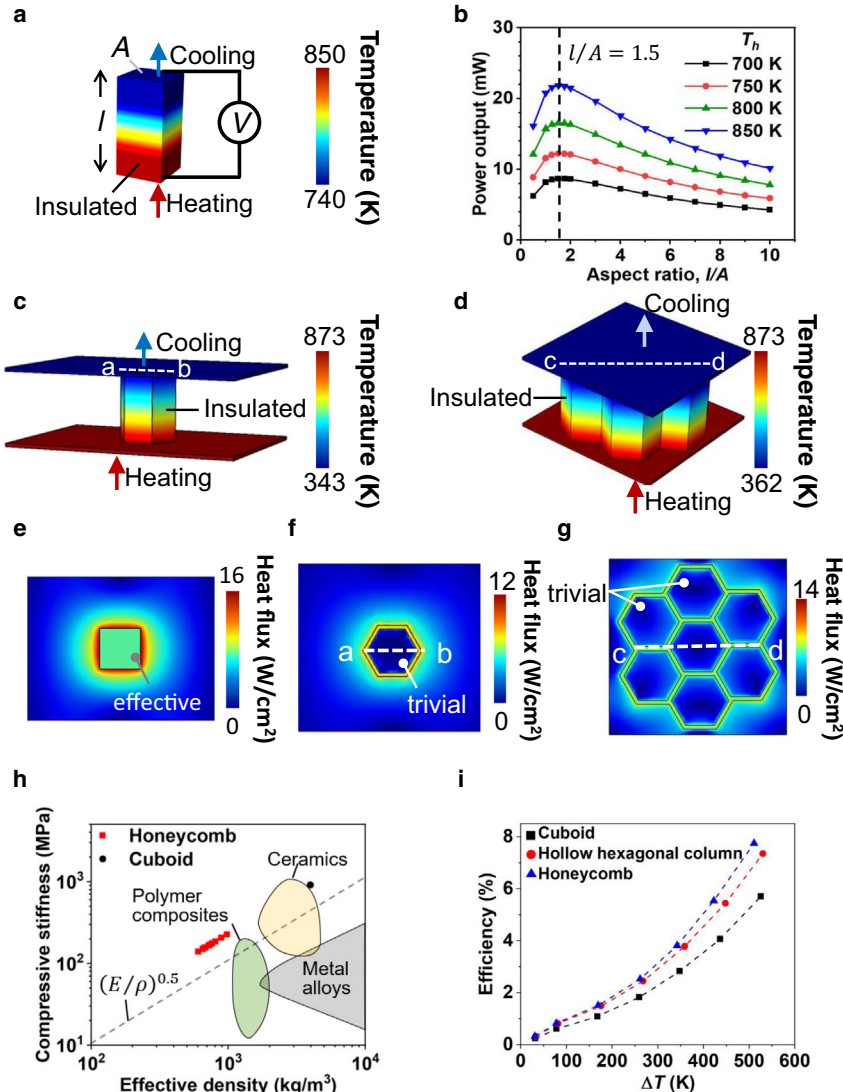

**Fig. 1 Design of Cu₂Se TE cellular architectures. a** Simulated temperature distributions in cuboid-, -shaped Cu₂Se TE legs. **b** Output power of cuboid-shaped Cu₂Se TE leg as a function of aspect ratio. **c, d** Simulated temperature distributions in (**c**) hollow hexagonal column-, and (**d**) honeycomb-shaped Cu₂Se TE legs. **e–g** Simulated heat flux distribution in the cold side of (**e**) cuboid-, (**f**) hollow hexagonal column-, and (**g**) honeycomb-shaped Cu₂Se TE legs when the hot side temperature is 873 K. **h** Compressive stiffness-density Ashby plot showing the cellular Cu₂Se materials described in this paper compared with other materials. **I** Generation efficiencies of Cu₂Se TE legs at the hot side temperatures ranging from 323 to 873 K.

stresses such as compressive or tensile stresses because of materials' thermal expansion or shear, torsion, and bending stresses arising from external environments. However, despite its actual importance, the module design for mechanical durability has been largely neglected so far. In this study, we propose a cellular honeycomb architecture to improve the mechanical durability of a TE leg. Honeycomb structures have been known to exhibit high strength to weight ratio, which comes from the efficient dispersion of external forces, particularly in the out-of-plane load[31]. Further, the in-plane compressive strength can be optimized by controlling the ratio between wall thickness and the unit-cell size. Considering our printing capability, we consider a cuboid, a hollow hexagonal column, which is a basic unit to build a honeycomb architecture, and a honeycomb comprising of seven hexagonal columns (Fig. 1c–g). Figure 1c, d show the cross-sectional areas of the cellular architectures. In our design, hexagon length is 2.5 mm, wall thickness is 0.33 mm, and the inclination angle is 30°. The cross-sectional area is 4 mm² for the hexagonal column and 105.21 mm² for the honeycomb, resulting in an aspect ratio of 1.5 for the hexagonal column and 0.06 for the

honeycomb. The leg length of all TEGs in this work is 6 mm. We hypothesize that the optimum aspect ratio is similar for both hexagonal column and cuboid TE legs, since the aspect ratio is a key factor for both thermal and electrical resistances. To accurately optimize the aspect ratio, the electrical and thermal contact resistances at TE leg-electrode interface need to be considered. When electrical current or heat transfers from the TE leg to an electrode, the cross-sectional shape of the TE leg affects how the current or heat spreads into the electrode[51]. Furthermore, solder properties may be influenced by the TE leg thickness, if the TE leg is not sufficiently thick. If empirical correlations between the TE leg geometry and contact resistances are available for the cellular or other architectures, the topology of the 3D TE module will be more accurately determined. For the honeycomb architecture, we select a relatively small aspect ratio, because it is challenging to achieve the optimum $l/A$ in the actual device, as fabricating high-aspect-ratio, multiple hexagonal columns are not feasible yet.

The mechanical properties of the 3D-printed Cu₂Se were measured by compressive test under the uniaxial compression

mode on a cuboid and a honeycomb. The stress-strain curve of the 3D-printed honeycomb in the elastic region exhibits similar behaviour to that of the cuboid (Supplementary Fig. 3). The calculated modulus 939 MPa, which was almost identical to 911 MPa of the cuboid. Interestingly, the honeycomb exhibited a larger plastic deformation region and higher fracture strain by three times, compared with the cuboid. This improvement may originate from unique structural characteristics of the honeycomb, which can distribute the stress concentration into a whole structure[52–54]. Sun et al.[54] demonstrated the distribution of the stress concentration in honeycomb architectures by the FEM simulations and experiments. Based on the measured properties, we predicted the compressive stiffness of the $Cu_2Se$-honeycomb architecture using the Ashby-Gibson relation (Fig. 1h). As the number of unit cell in the honeycomb architecture increases, effective compressive stiffness decreases in the range of 100–200 MPa due to the decrease in density ($\rho$), agreeing with the measured value of 174.7 MPa in the honeycomb with the effective cross-sectional area including pores. The effective density of the $Cu_2Se$-honeycomb architecture is merely 600–1000 kg m$^{-3}$ Compared with other common materials with a density near 1000 kg m$^{-3}$, the $Cu_2Se$-honeycomb architecture exhibits a superior property. Other materials with high specific stiffness follow $(E/\rho)^{0.5}$ line in the Ashby $E$-$\rho$ plot. However, the honeycomb architecture shows the specific stiffness of 198 kPa m$^3$ kg$^{-1}$ that is greater than $(E/\rho)^{0.5}$ by a factor of 2. The honeycomb architectures take <25% density of the traditional cuboid architecture, making it lightweight and stiff inorganic TE materials. Thus, for mobility applications and for large-scale energy harvesting systems, the hexagonal architecture can be considered useful due to its excellent strength-weight ratio.

In addition to the mechanical properties, the simulation results showed that the proposed cellular architectures could enhance the power generation efficiency, compared with a cuboid. With 3D FEM, we compared three different topologies including a cuboid, a hollow hexagonal column, and a honeycomb. The generation efficiency is defined as the ratio of power output to heat input. A hexagonal column and honeycomb architectures are expected to exhibit 7.3% and 7.7% efficiency when the hot-side temperature is 873 K that are 22 and 26% greater than the efficiency of traditional cuboids (Fig. 1i). Under similar $\Delta T$, both the hexagonal column and cuboid TEGs with an equal aspect ratio produce analogous potential distributions (Fig. 1a, c, and d). However, compared with a cuboid module, in hexagonal and honeycomb modules, less amount of heat is required to achieve the same $\Delta T$ than the cuboid module due to less effective heat spread in the electrodes. In the hexagonal module, the TE legs contact electrodes over a large region enclosing an area of 16.24 mm$^2$. As seen in the simulated heat flux distribution (Fig. 1e–g), the steady-state heat flux is predicted as trivial within the enclosed area on the cold side of the hexagonal and honeycomb modules because the electrode temperature is maintained uniform (Fig. 1e). The uniform temperature within the enclosed region strongly suppresses in-plane heat flow within the electrode. However, the cuboid module efficiently spreads heat from the electrode centre to outward radially (Supplementary Fig. 4), leading to reduced TE leg-electrode interface thermal resistance. The inefficient heat spreads in electrodes cause the hexagonal and honeycomb modules to achieve a large generation efficiency. Interestingly, the efficiency slightly increases when the honeycomb module scales up.

**Rheological design of $Cu_2Se$ TE inks**. To realize the designed topology of $Cu_2Se$ TE legs, we developed viscoelastic $Cu_2Se$ 3D printing inks, tailored by inorganic $Se_8^{2-}$ polyanion. The appropriate viscoelastic properties of colloid inks are crucial to ensure a fine flow through nozzles and the structural integrity of printed architectures. Typically, the addition of organic binders into inks can easily achieve their desired viscoelastic properties. However, these organic binders can act as impurities that degrade the electrical properties of printed objects due to the poor electrical interconnection among grains[44,45,47–49]. Our group reported the 3D printing process using organic binder-free all-inorganic $Bi_2Te_3$-based TE inks, tailored by the inorganic anions of $Sb_2Te_4^{2-}$ ionic chalcogenidometallate (ChaM) molecule[55–57]. This inorganic anion is beneficial for securing the viscoelasticity of inks and producing purely inorganic 3D-printed objects with high $ZT$ values, which is the evaluation of energy conversion efficiency in materials by the equation of $ZT = S^2\sigma T/\kappa$, where $S$, $\sigma$, $\kappa$, and $T$ are the Seebeck coefficient, electrical conductivity, thermal conductivity, and absolute temperature, respectively. The viscoelasticity of current organic binder-free $Cu_2Se$ TE inks was achieved through the use of inorganic $Se_8^{2-}$ polyanion that effectively hold $Cu_2Se$ particles together in an electrostatic manner. Based on the viscoelasticity, we can create complex architectures of TE materials through the 3D printing process (Fig. 2a and Supplementary Movie 1).

For the rheological design of $Cu_2Se$ TE inks, the contents of inorganic $Se_8^{2-}$ polyanion were controllably varied, and their properties were assessed in terms of various rheological property measurements. Figure 2b shows the dynamic viscosity ($\eta'$) curves of various $Cu_2Se$-based inks as a function of the ion content. The pristine $Cu_2Se$-based ink behaves like a Newtonian fluid with a very low $\eta'$. As the ion was incorporated in the ink, an increase in $\eta'$ was observed, and the inks began to exhibit strong dependency on the shear rate, whose behaviour is generally called as a Bingham fluid. In the previous literature, we have reported the role of ChaM as a rheological modifier for all-inorganic BiSbTe inks[55], but its efficiency appears to be much more pronounced in $Cu_2Se$ inks. In theory, a Bingham fluid reflects the presence of a 3D network structure of a colloidal system that experiences the structural collapse under shear stress, and so a sudden viscosity decrease is observed[58,59]. Thus, a stronger shear rate dependence and stronger pseudo-network structure formation were observed with increasing the content of polyanion.

In order to obtain a continuous discharging during 3D printing, stable fluidity is one of the essential requirements. As shown in the stress sweep results (Fig. 2c), all the inks undergo the sudden drop of $\eta'$ at a certain stress, which is indicative of the phase conversion from linear and nonlinear viscoelastic flows[60,61]. Particularly, the inks with a $Se_8^{2-}$ polyanion of 80 and 110 wt% give an overshoot at the transition, a sign of phase instability of those inks under high shear stress. This finding suggests that the excessive $Se_8^{2-}$ polyanion can bring about plugging nozzle and discontinuous discharging. In addition, the inks would experience different shear stresses during printing that can vary the viscoelastic properties of them. As a result, the thixotropic structural recovery may also be affected. Figure 2d shows the variation of the storage modulus (G′) at 1 rad s$^{-1}$ before and after applying high shear stress and G′ ratio (G′after/G′before) that implies the degree of thixotropy from the frequency sweep test[62]. The G′ value of inks except for the $Se_8^{2-}$ fraction of 20 wt% exhibited an almost complete recovery to the original value after experiencing the shear stress. Even though G′after/G′before of the pristine ink is near to 1, the low viscosity makes it difficult to be printed. The variation of rheological properties as a function of the ion also shows that the limited interaction between the $Cu_2Se$ particles can be strengthened by incorporating the ions. In this regard, the polyanion fraction of 20 wt% is insufficient to obtain structural integrity. The thixotropic behaviour as a function of the shear stress of the pristine ink and ink containing 50 wt% ions was also evaluated using three-

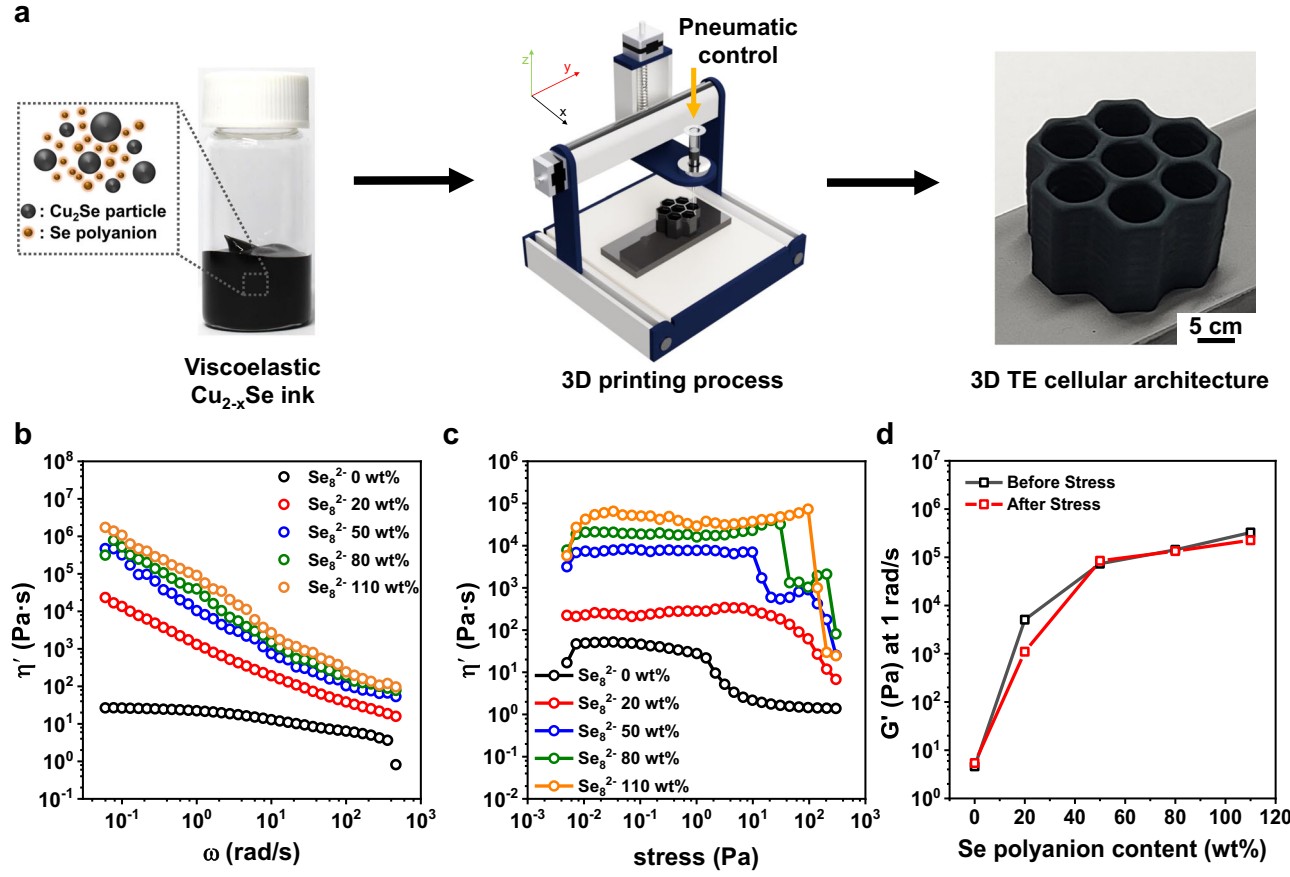

**Fig. 2 Optimized viscoelasticity of Cu$_2$Se-based ink. a** Scheme for 3D printing process of the Cu$_2$Se-based honeycomb cellular architecture by using all-inorganic Cu$_{2-x}$Se ink. **b** The dynamic viscosity ($\eta'$) curves of all-inorganic Cu$_2$Se TE inks from the frequency sweep test. **c** Stress sweep results of the $\eta'$ over the shear stress range of 0.005 300 Pa. **d** The variation of storage modulus (G') and G' at 1 rad s$^{-1}$ as a function of the Se$_8{}^{2-}$ polyanion content.

interval thixotropy tests (Supplementary Fig. 5). From the measurement, the polyanion-containing ink exhibits two distinctive features: (1) rapid recovery rate after the stress, (2) nearly identical value of G' after full recovery compared to the pristine ink, and (3) enhanced ink dispersion stability (Supplementary Fig. 6). These findings suggest that the 50 wt% polyanion containing ink has the optimal printing capability.

**3D printing of Cu$_2$Se TE materials**. Owing to the desirable viscoelasticity of the Cu$_2$Se ink containing 50 wt% of the Se$_8{}^{2-}$ polyanion, various 3D shapes of Cu$_2$Se TE materials were built by the 3D printing process (Fig. 3a and Supplementary Fig. 7). The optical microscopy (OM) image reveals that the printed layers were highly uniform with a single-layer thickness of 330–340 μm (Fig. 3b). The printed samples were further sintered at 873 K for the desired duration. During the sintering, the Se$_8{}^{2-}$ polyanion aided the densification of the 3D-printed Cu$_2$Se to produce robust TE materials with the liquid-phase sintering effect. During the heat treatment of the 3D-printed samples, the Se$_8{}^{2-}$ polyanion was initially decomposed to form crystalline Se among Cu$_2$Se particles, which was subsequently liquefied as manifested in the differential scanning calorimetry result (Supplementary Fig. 8). The liquefied Se filled up the pores and voids among Cu$_2$Se particles through the capillary force, promoting uniform sintering of Cu$_2$Se particles at high temperatures. The scanning electron microscopy (SEM) images show that the Cu$_2$Se particles were mutually merged into the polycrystalline phase, as shown in the comparison of the before and after heat treatment process (Fig. 3c, d). The X-ray diffraction patterns reveal that the sintered

samples consisted of two different phases of Cu$_{1.8}$Se and Cu$_2$Se (Fig. 3e). With a longer sintering time, the peaks corresponding to those of Cu$_{1.8}$Se were progressively diminished, whereas the peaks for Cu$_2$Se were pronounced. This result can be attributed to the evaporation of excess Se during the sintering. This finding was further supported by the thermogravimetric analysis of the dried Cu$_2$Se inks, in which a significant weight loss was observed at 600–700 K, agreeing with the temperature of Se evaporation (Supplementary Fig. 9)[63]. In addition, despite the sintering shrinkage of 60–70%, the sintered materials well maintained their primary architectures without substantial distortion (inset of Fig. 3d), allowing us to design the dimensions and shapes of TE materials with the pre-computer-aided design.

**TE properties of 3D-printed Cu$_2$Se materials**. This compositional modulation with various sintering conditions is beneficial in controlling over the electrical properties of printed materials because the intrinsic defects of Cu vacancy in Cu$_2$Se act as a hole donor. As expected, the room-temperature carrier concentrations of the samples measured by the Hall effect measurement decreased from $6.68 \times 10^{20}$ to $2.27 \times 10^{20}$ cm$^{-3}$ with the duration time increasing from 1 to 7 h at 873 K (Fig. 4a). In addition, the hole mobility ($\mu_h$) increased from 14.13 to 33.67 cm$^2$ V$^{-1}$ s$^{-1}$ with an increase in sintering time of ~1–3 h, and it decreased to 23.82 cm$^2$ V$^{-1}$ s$^{-1}$ as the sintering time increased to 7 h. The increase in the hole mobility can be understood with the consideration of the decrease in the defect density. At the same time, the sample density decreased with the increase in the sintering time (Supplementary Table 1), which comes from the Se

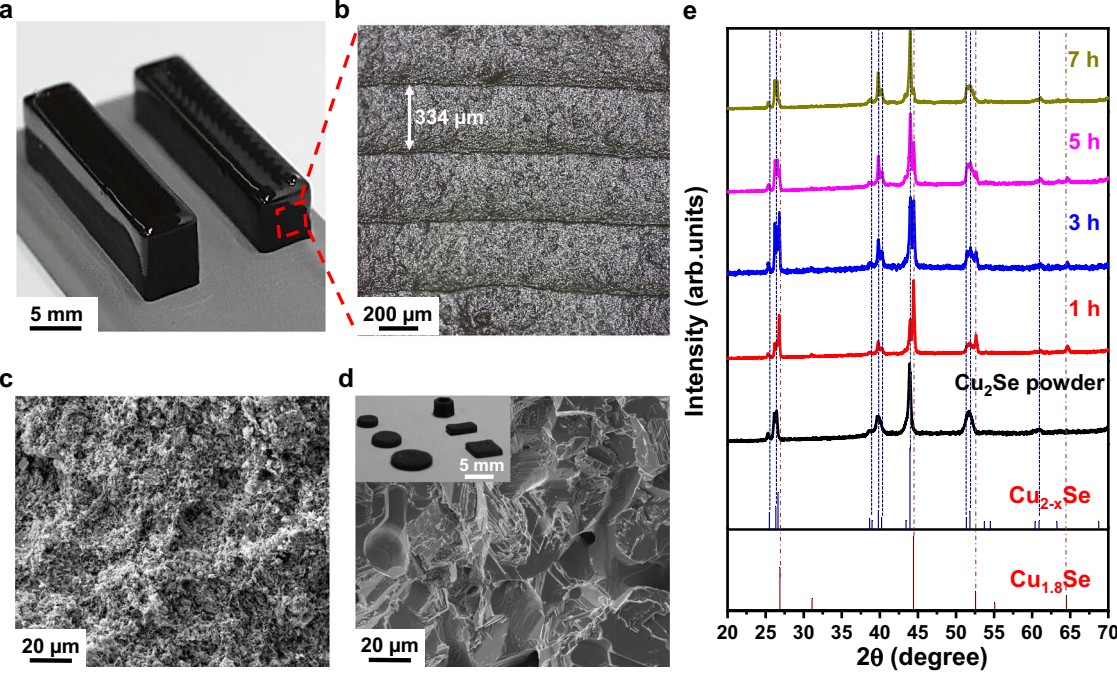

**Fig. 3 3D printing of Cu₂Se TE materials. a** Photograph and **b** OM image of the 3D-printed Cu₂₋ₓSe cuboids. **c, d** SEM images of fractured surfaces of the 3D-printed Cu₂₋ₓSe (**c**) before and (**d**) after sintering at 873 K for 5 h. The inset of the panel (**d**) shows the photograph of 3D-printed and sintered Cu₂Se TE materials with various samples. **e** XRD patterns of as-synthesized Cu₂Se powder and sintered samples for 1, 3, 5, and 7 h for 873 K.

evaporation. The decrease in the density of the samples may contribute to the decrease in the hole mobility of the samples sintered for longer duration times. Although the sample density and porosity were slightly affected by the sintering conditions, the SEM images show that all the samples were well sintered, and substantial changes in the microstructures were not observed (Supplementary Fig. 10).

The major challenges in the 3D printing of TE materials are homogeneity in TE properties with respect to dimensions and shapes. Although Cu₂Se is known to have the isotropic TE properties owing to the isotropic crystal structure of the face-centred cubic above 100–150 °C, the inhomogeneity of the TE properties can be caused by the inhomogeneity of the distribution of temperatures and particle density in the dried specimen during the sintering. We characterized the room-temperature electrical conductivity and Seebeck coefficient of four different samples of a disc, two cuboids, and a hollow hexagonal column. As shown in Fig. 4b, all the samples exhibited identical electrical conductivities and Seebeck coefficients within the error range of the measurement (10%), demonstrating the homogeneity in the TE properties of the 3D-printed Cu₂Se materials.

The temperature-dependent TE properties of the 3D-printed cuboid-shaped Cu₂Se sintered at 873 K for 1, 3, 5, and 7 h were characterized at temperatures from 300 to 1000 K. All properties are in agreements with the typical temperature dependences of the reported Cu₂Se polycrystals, where the electrical and thermal conductivity is in negative dependence on the temperatures and vice versa for the Seebeck coefficient (Fig. 4c–e). At room temperature, the highest electrical conductivity of 150,000 S m⁻¹ was observed at the 1 h-sintered sample and it gradually decreased to 31,000 S m⁻¹ at 1000 K. In addition, the highest Seebeck coefficient of 186 μV K⁻¹ at 1000 K was exhibited by the 7 h-sintered samples. These values agree with the reported values of Cu₂Se polycrystals[12]. Moreover, the fluctuations of the electrical and thermal properties were clearly observed at temperatures ranging from 350 to 400 K. This phenomenon is

attributed to the phase transition of the α phase into the β phase of Cu₂Se, consistent with the reported temperature ranges[64,65]. The sintering temperature dependency of the properties was clearly observed. For example, the electrical conductivities increased, and the Seebeck coefficients decreased with decreasing sintering duration time in the entire measurement temperatures, agreeing with the changes in the hole concentrations since the electrical conductivity and Seebeck coefficient have the trade-off relation with the variable of the carrier concentrations[56]. At the optimum condition of the sintering duration time of 5 h, the samples exhibited the highest power factor of 6.3 μW cm⁻¹ K² at 1000 K (Supplementary Fig. 11).

The thermal conductivity decreased with longer duration times for sintering (Fig. 4e), which can be attributed to the higher porosity observed in the sintered samples for longer duration times, acting as scattering sites of phonons. The minimum thermal conductivity was found to be 0.5 W m⁻¹ K⁻¹ for the samples sintered for 7 h at 1000 K, which was lower than the reported values for bulk Cu₂Se polycrystals[12–14]. To understand this low thermal conductivity, the lattice thermal conductivities (κ_L) of the samples were calculated by subtracting the electronic thermal conductivity (κ_e) from the total thermal conductivity (Supplementary Fig. 12). κ_e was calculated by the Wiedermann–Franz relationship ($\kappa_e = L\sigma T$), where L is the Lorenz number. The calculated lattice thermal conductivity showed similar trends to the total thermal conductivity, and a minimum κ_L of 0.16 W m⁻¹ K⁻¹ was observed in the 7 h-sintered Cu₂Se samples. These very low values can be understood by taking into account the multi-scale porosity that the 3D-printed samples have a lower density than the bulk values and can consequently scatter phonons over a wide range of wavelengths. We further calculated the κ_L of our samples with 100% density through the modified effective-medium theory[66]. The calculated lattice thermal conductivities of the 5 and 7 h-sintered samples (Supplementary Fig. 13) ranged from 0.35 to 0.5 W m⁻¹ K⁻¹, consistent with the range of the reported values[12–14].

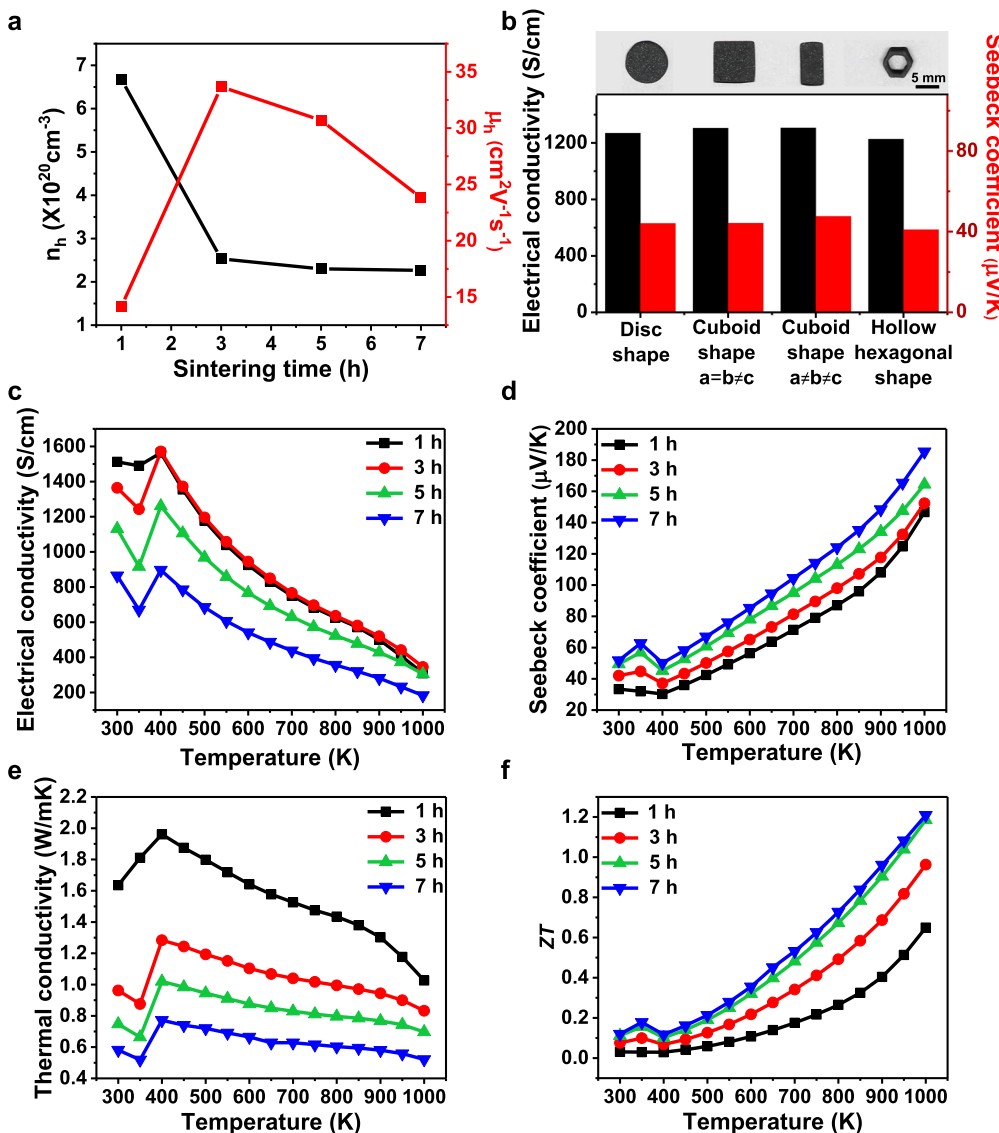

**Fig. 4 TE properties of the 3D-printed Cu₂Se samples. a** Room-temperature hole concentrations and mobilities of the Cu₂Se samples sintered at 1, 3, 5, and 7 h at 873 K. **b** Room temperature electrical conductivities and Seebeck coefficients of various shapes of Cu₂₋ₓSe samples. **c–f** Temperature-dependent (**c**) electrical conductivities, **d** Seebeck coefficients, **e** thermal conductivities and **f** ZT values the Cu₂Se samples sintered at 1, 3, 5, and 7 h at 873 K.

Consequently, the maximum *ZT* value was achieved by the sample sintered for 7 h, marking 1.21 at 1000 K (Fig. 4f). This value is slightly lower than or similar to the reported bulk value of the undoped Cu₂Se[12]. This maximum value is one of the highest among the reported TE materials prepared from various inks or by 3D printing process (Supplementary Table 2). Compared with the recently reported TE foams, the *ZT* value of our 3D-printed Cu₂Se is at least an order of magnitude higher than those, as summarized in Supplementary Table 3.

**Fabrication and evaluation of 3D-printed Cu₂Se TE modules.** To validate our designs of nature-inspired cellular architectures, we prepared three different TE legs of a cuboid, a hollow hexagonal column, and a honeycomb and fabricated the p-type single leg-power generators by the 3D printing process (Fig. 5a–c). Recently, a few studies have been reported on n-type semiconductors of ternary or quaternary copper chalcogenide compounds of CuAgSe[67], CuFeS₂, and Cu₁₋ₓZnₓFeS₂[68]. However, the *ZT* values of these materials are <0.6 at the highest, not high enough to use them as an n-type pair of our Cu₂Se. As alternatives, different classes of n-type

TE semiconductors such as doped SnSe, skutterudites, half-Heusler can be potential candidates as an n-type cellular pair of our 3D-printed Cu₂Se. The top and bottom of the 3D-printed Cu₂Se TE legs were metallized with Ni layers through sputtering, subsequently attaching to Cu electrodes using an Ag paste as a solder. For a reliable measurement, the area ratios of the cold-side Cu electrodes/ TE legs were equalized for all modules to ensure an equivalent cooling rate. Also, we set the identical hot-side temperatures for the measurement of the power-generating performances of these modules by heating the top of the modules with a ceramic heater. The cold side was cooled at the bottom with a water-circulating cooler under the same water flow rate and water temperature (Supplementary Fig. 14). Upon heating, all modules showed almost linear increases in the output voltages and quadratic increases in the output power, demonstrating the reliability of the measurement (Fig. 5d–f). Moreover, the measured electrical output values and of the TEGs are in line with the simulation results (Fig. 5g). This agreement suggests that the simulated models (Fig. 1i), that predicted higher efficiencies of the hexagonal- and honeycomb modules than that of the cuboid module, were well realized in the 3D-printed modules.

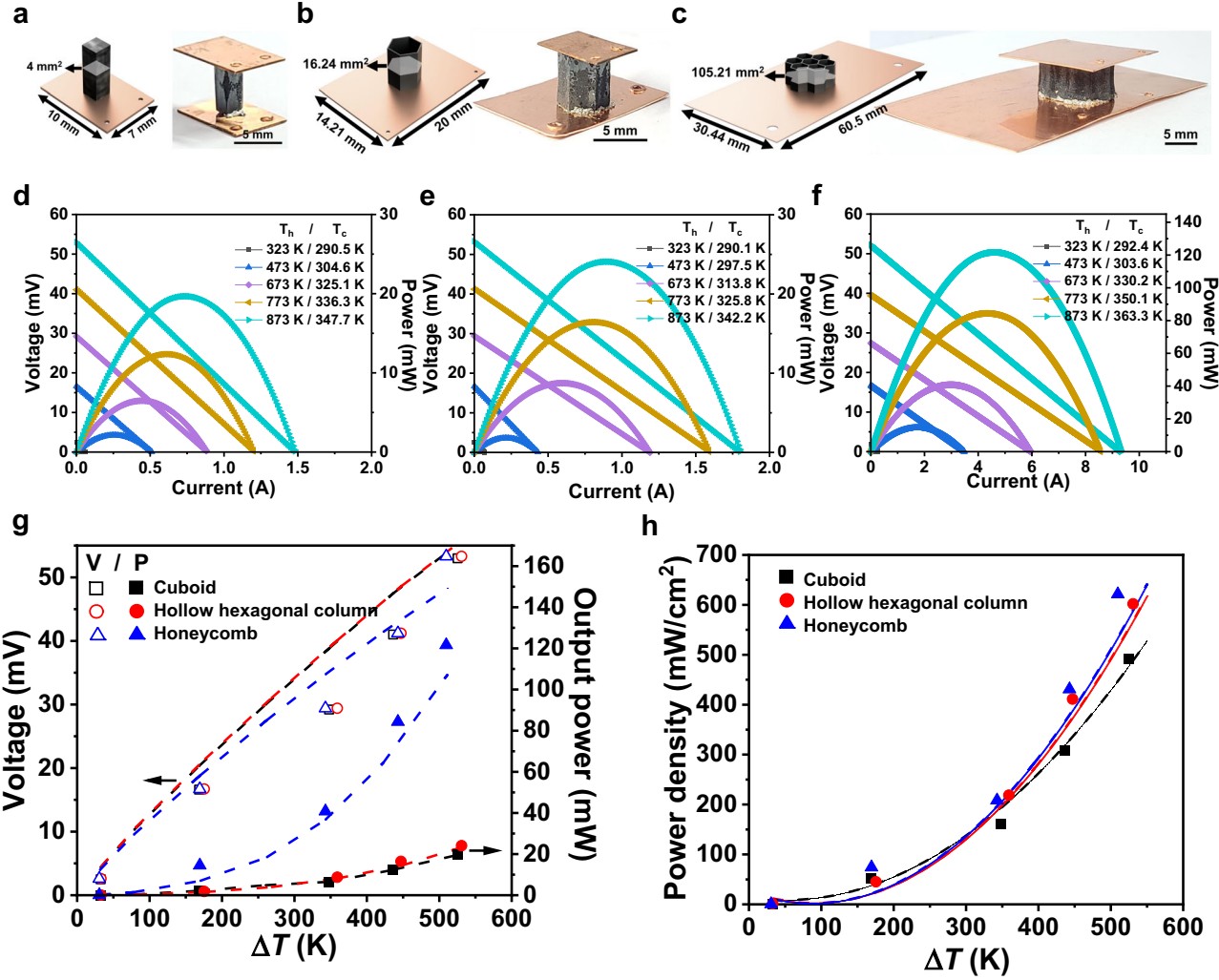

**Fig. 5 power-generating TEG with 3D-printed cellular honeycomb material. a–c** 3D illustrated models and photograph of TEGs chipped with (**a**) cuboid-, (**b**) hollow hexagonal column-, and (**c**) honeycomb-shaped $Cu_2Se$ TE legs. **d–f** Output voltages and powers of (**d**) cuboid-, (**e**) hollow hexagonal column-, and (**f**) honeycomb-shaped $Cu_2Se$ TE legs at the hot side temperatures ranging from 323 to 873 K. **g** Output voltages and powers, and **h** power densities of TEGs as a function of the temperature differences. The points and the dotted lines in the panels of (**g**) and (**h**) are the measured values and the predicted values by the FEM simulation.

In addition, the honeycomb-based module showed a significantly larger power than those of the other modules (Fig. 5g), which can be understood by a lower module resistance due to its larger cross-sectional area by approximately five times. This larger cross section is beneficial in the reduction of the contact resistance at TE legs and electrodes. The calculated contact resistances of the modules from the material properties were the lowest at the honeycomb among all the modules. Accordingly, the honeycomb exhibited the highest power density of 621.40 mW $cm^{-2}$ among the three topologies (Fig. 5h). These power-generating performances validate our topology designs and demonstrates feasibility of the proposed strategy of the topological designs of TE legs for enhancing power performance and mechanical durability.

To further demonstrate the working stability of the 3D-printed $Cu_2Se$ material, we characterized the electrical conductivity and Seebeck coefficient through multiple heat cycles in the temperature range from room temperature to 873 K, which was the highest hot-side temperature of the module for the power generation measurement. In the cycle test, the electrical conductivities and Seebeck coefficients were well preserved without degradation within the equipment error range by three

times (Supplementary Fig. 15). Moreover, we tested the working stability of the 3D-printed samples for a longer duration time by measuring their TE properties during the thermal annealing at 873 K. As shown in Supplementary Fig. 16, the 3D-printed $Cu_2Se$ sample maintained both the primary electrical conductivity and the Seebeck coefficient for 1 h without thermal degradation. These results clearly demonstrated the working stability of our 3D-printed $Cu_2Se$ for the power generation application.

## Discussion

In summary, we have demonstrated the applicability of 3D-printed cellular architectures of $Cu_2Se$-based TE materials for designing efficient and durable TEGs. We developed the 3D FEMs for cuboid, hollow hexagonal column, and cellular honeycomb structures that allowed us to comparatively design the topology of TE legs with perspectives of higher output power and mechanical strength. Moreover, we developed all-inorganic $Cu_2Se$ TE inks with tailored rheological properties by the electroviscous effect of $Se_8^{2-}$ polyanions. These inks allowed the building of complex 3D architectures of $Cu_2Se$ TE materials that exhibit a peak ZT value of 1.21 at 1000 K. The experimental measurements of the fabricated TEGs chipped with the 3D-printed TE legs

showed good agreement with the predicted power-generating performances, in which the honeycomb TE architecture with the highest predicted mechanical stiffness exhibited superior power density to typical cuboid TE legs. Undoubtedly, higher power generation performance can be obtained in the future using TE materials with higher $ZT$ values. In addition to improving material properties, this study shows another system-level design to enhance the power-generating performance and durability in TE modules through the topological design of TE legs, which will accelerate the application of TE power generators to prevalent fields. Furthermore, our 3D printing approach has great application potential for the cost-effective manufacturing of well-designed TE modules, which can be easily transferred to other fields of electronic and energy devices.

Recently, the TE foams constructed with various classes of materials such as oxides, organics, carbon allotropes, and their hybrids have been reported. Although the reported TE foams might also be categorized into the cellular TE materials in view of their porosities, they are clearly distinct from our cellular honeycomb architecture in terms of (i) the purpose and benefit of the structuring, (ii) pore structures, (iii) fabrication process, and (iv) TE properties (Supplementary Table 3). Especially, most reports of the TE foams generally aim for the enhancement of $ZT$ values in materials by the formation of stochastic pores to intensify phonon scattering, which can reduce the thermal conductivity of materials. However, in this study, we demonstrated that our designs with periodic pore structures, manufactured by the 3D printing process can optimize the system-level thermal transport and structure-induced mechanical stiffness, improving the efficiency and durability of modules. To further optimize the topological design for the TE architecture in a large design space, topology optimization may be considered. The topology optimization identifies non-intuitive and mathematically optimal structures, and has been employed widely for mechanical structural systems. However, conventional topology optimization requires extensive computation resources, and often results in complex designs that cannot be fabricated by legacy techniques. Thus, topology optimization is not yet considered as widely applicable approach in the field.

## Methods

**Materials**. Ethylenediamine (>99.5%), ethanethiol (>97%), and glycerol (>99.5%) were purchased from Aldrich Chemical Co. Powders of Cu (99.9%) for high-energy ball milling, Se (99.999%) for the synthesis of Se polyanion were purchased from Alfa Aesar. Powder of Se (99.999%) for high-energy ball milling was purchased from 5 N Plus. Isopropanol was purchased from Samchun Chemicals. The chemicals and elements were used without purification.

**Synthesis of all-inorganic Cu$_{2-x}$Se-based ink**. The entire process was accomplished under inert condition. The Cu$_2$Se powder was fabricated by high-energy ball milling (SPEX, 8000 M Mixer/Mill) of Cu and Se powder with the stoichiometric composition of Cu$_2$Se for 200 min. The ball-milled powder was sieved for removing particles larger than 45 μm. The soluble ionic Se binder was synthesized by dissolving 0.5 g of Se powder in the co-solvent of 0.5 mL ethanethiol and 4.5 mL ethylenediamine with vigorous stirring at room temperature[69]. After 72 h, the dissolved solution was added 37.5 mL of isopropanol and centrifuged at 7715 × $g$ for 10 min to precipitate Se ion binder. The precipitates, discarded supernatant, was dried for 1 h under vacuum condition. The all-inorganic Cu$_{2-x}$Se ink was prepared by mixing the 2.5 g of glycerol, 2 g of the ball-milled Cu$_2$Se powder, and 1 g of dried Se ion binder with a planetary centrifugal mixer (ARM-100, Thinky) for 2 h to fully disperse. In the mixing process, six zirconium oxide grinding balls with 5 mm diameter were added for homogenizing, effectively.

**Measurement for rheological properties of the Cu$_{2-x}$Se inorganic ink**. The rheological properties of Cu$_{2-x}$Se inks were analysed using a rotational rheometer (Haake MARS III, Thermo Scientific) equipped with a coaxial cylinder geometry at room temperature. The frequency sweep tests were carried out at a constant stress of 1 Pa and the stress sweep tests were conducted over the range of 0.005–300 Pa at a frequency of 1 rad/s. The three-interval thixotropy tests(3ITT) at various stresses (1, 5, 10, 50, and 200 Pa) were also conducted as reported elsewhere[62,70,71].

**3D printing and heat treatment**. Three-dimensional printing was carried out with a home-built extrusion-based 3D printer and programmable control of temperature and pressure. The synthesized ink was loaded in a 5 mL syringe (Saejong) which had a metal nozzle with an inner diameter of 340 μm. By the design software, the ink was printed with layers of parallel lines and developed perpendicular to the layers (Supplementary Movie 1). The printing process was performed at room temperature and has intervals of 1 s. As-printed sample was dried at 423 K for 5 h under inert condition, then sintered with two-step heat treatment process at 623 K for 1 h, and 873 K for 1–7 h under mixture atmosphere with 96% of N$_2$ and 4% of H$_2$.

**Characterization of 3D-printed samples**. The OM image was acquired with an Olympus BX51M. The scanning electron microscope images were acquired with a field-effect SEM (Nova-NanoSEM230, FEI and S-4800 Hitachi High-Technologies) operated at 30 kV. The XRD patterns were obtained by Rigaku D/Max2500 V diffractometer equipped with a Cu-rotating anode X-ray source ($\lambda = 0.15418$ nm), operating at 40 kV and 30 mA. The reference peaks of Cu$_{2-x}$Se, Cu$_{1.8}$Se are corresponding to JCPDS: 00-047-1448 and 01-073-8642, respectively.) Thermogravimetric analysis of dried ink and 7 h sintered sample was carried out using a TA-Q500 thermal analyzer at a heating rate of 10 K min$^{-1}$ under nitrogen condition. Compression test at room temperature was performed under uniaxial loading with a strain rate of $1 \times 10^{-3}$ s$^{-1}$ by Instron-5948 equipped with a 2 kN load cell.

**TE properties of 3D-printed samples**. Temperature-dependent electrical conductivity and Seebeck coefficient were characterized with a commercial equipment (SBA 458 Nemesis, Netzsch) in temperature range from 300 to 1000 K under argon atmosphere. Thermal conductivity in the same temperature range of electrical conductivity was calculated by the equation $\kappa = \rho C_p D$, where $\kappa$ is the thermal conductivity, $\rho$ is the density, $C_p$ is the specific heat capacity and D is the thermal diffusivity. The density of the 3D-printed materials was obtained by measuring their volume and weight. The specific heat capacity was calculated by Dulong-Petit equation and the thermal diffusivity was measured with laser flash analysis (LFA 467HT, Netzsch) in the same temperature range of electrical conductivity. The equipment error range of thermal diffusivity by the radiation heat loss is within 0.35% at maximum[72]. The estimation of lattice thermal conductivities of 100% dense samples by the modified formulation of effective medium theory as follows:

$$\kappa_L = \kappa_h \frac{(2 - 2\Phi)}{(2 + \Phi)} \tag{1}$$

suggested by Lee et al.[70], where $\kappa_h$ and $\Phi$ are the lattice thermal conductivity of host materials and the porosity, respectively. The room temperature electrical conductivities were measured by Van der Pauw 4-point method (Keithley 2400 source-meter controlled by Lab trace 2.0 software, Keithley Instrument, Inc). The room temperature Seebeck coefficients were characterized with the open-circuit voltage and temperature gradient by two commercial Peltier coolers, contacted with the 3D-printed samples. The voltage and temperature difference of the sample were measured with T-type thermocouples connected by Keithley 2000 multimeter. The Seebeck coefficient was calculated from the slope of the voltage-temperature difference curve, consisting of four data points by different temperature gradient from $-5$ to $+5$ K. This home-built equipment has accuracy with an error range of 3% by confirming the n-type Bi$_2$Te$_3$ and p-type BiSbTe ingot samples. The room temperature carrier concentrations and mobilities were measured by a Hall measurement equipment (HMS-5000, ECOPIA) with the magnetic field of 0.55 T.

**Simulation of thermoelectric device properties**. A three-dimensional, steady-state FEM was developed using a commercial software, COMSOL Multiphysics, to calculate the temperature distribution in a TEG and resulting electrical outputs. In the FEM, the size and shape of TEGs were identical to the actual 3D-printed devices as illustrated in Fig. 5a–c. The model employed experimentally obtained, temperature-dependent properties of 3D-printed Cu$_2$Se as shown in Fig. 4. As boundary conditions, the hot-side temperature was fixed to experimentally acquired values shown in Fig. 5d–f while the cold-side electrode was subject to forced convection. A convection coefficient at the cold-side surface was selected as 200 W m$^{-2}$ K$^{-1}$, as the simulated cold-side temperatures matched with the experimental measurements by using this convection coefficient. Similar to the experimental condition, the cooling water temperature was set to 290 K. The lateral side of the TEG was adiabatic. The cold-side electrode was electrically grounded. Supplementary Fig. 2 depict the schematic of thermoelectric FEM showing the distributions of simulated electrical potential when the hot-side temperature was 873 K.

**Calculation of effective specific modulus**. First, Ashby and Gibson relation for hexagonal lattices was used to calculate the effective modulus of the honeycomb architecture in the out-of-plane direction ($E$). Then, $E$ was divided by an effective density ($\rho$) to obtain effective specific modulus. For Ashby and Gibson relation, we define the wall thickness ($t$), hexagon length ($l_h$), and inclination angle ($\theta$). Considering the actual measurements, we use $t = 0.33$ mm, $l_h = 2.5$ mm, $\theta = 30°$, actual density ($\rho_s$) = 3970 kg m$^{-3}$, and actual Young's modulus of 3D-printed Cu$_2$Se ($Es$) = 911 MPa.

Relative density

$$\rho = \left(\frac{t}{l_h}\right)\frac{3}{2(1+\sin\theta)\cos\theta}\rho_s \tag{2}$$

Modulus

$$E = \left(\frac{t}{l_h}\right)\frac{3}{2(1+\sin\theta)\cos\theta}E_s = \frac{\rho}{\rho_s}E_s \tag{3}$$

**Fabrication and power measurement of Cu$_2$Se TEGs.** The cold-side Cu electrodes with the size specified in Fig. 5a–c and thickness of 0.3 mm were used to equalize the cooling rate of the three TEGs by matched the ratio between coverage area of 3D-printed samples and the area of the cold-side electrode was made same. The top and bottom of the 3D-printed Cu$_2$Se TE legs were metallized with 300 nm-thick Ni layers by the sputtering. And then, the samples were integrated with the Cu electrodes using Ag paste (Pyro-Duct 597-A, Aremco). The output power was measured with the ceramic heater (70 mm × 15 mm) as a heat source and water-circulating cooler. To support the hot plate, which was larger than fabricated TEG and prevented from thermal convection, the hot plate was wrapped with glass fabric. Temperature differences were measured with K-type thermocouples connected by the Keithley 2000 multimeter. The whole measurement was performed in a vacuum chamber to prevent from undesired oxidation.

## Data availability
Source data for figures are provided with this paper. The data that support the plots within this paper and other findings of this study are available from the corresponding author upon reasonable request. Source data are provided with this paper.

## Code availability
The COMSOL Multiphysics codes used in this study are outlined in the method section and available from the corresponding authors upon reasonable request.

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

## Acknowledgements

This work was supported by the Samsung Research Funding Center of Samsung Electronics under Project No. SRFC-MA1801-05. J.-Y.K. acknowledges the Research Fund (1.210035.01) of UNIST (Ulsan National Institute of Science and Technology).

## Author contributions

S.C. and F.E. contributed equally to this work. S.C., F.E., B.K., H.G.C., and J.S.S. designed the experiments, analyzed the data, and wrote the paper. S.C., F.K, J.L., S.E.Y., S.J., S.B., S.C. and K.K. carried out the synthesis and basic characterization of materials. H.J. and H.G.C. performed the characterization of rheological properties. G.K. and S.A. performed the characterization of thermal conductivities. H.K. and J.-Y.K. performed the characterization of mechanical properties. S.C. carried out the fabrication and measurement of TEGs. F.E. and B.K. performed the simulation studies. All authors discussed the results and commented on the manuscript.

## Competing interests

The authors declare no competing interests.
