## [Peer Review File · Nature Communications]

Editorial Note: Parts of this peer review file have been redacted as indicated to remove third-party material where no permission to publish could be obtained.

REVIEWER COMMENTS

Reviewer #1 (Remarks to the Author):

This manuscript reported the extrusion-based 3D printing fabrication of Cu₂Se-based thermoelectric cellular architectures for power generation application. The cellular design is interesting. However, the manuscript is far from the Nature Communications quality and is more suitable for a specialized journal.

First, the background and prior works in the field of cellular thermoelectric materials are poorly reviewed. The authors just simply pointed out the features of cellular designs for some areas without providing specific cons and pros of these designs. To date, some related reports on the thermoelectric foams have been published and what is the difference between the above works and the author's work? The authors should give a more insightful explanation in the manuscript.

Second, the reported thermoelectric material is common and the performance does not represent a significant advance when compared with state-of-the-art thermoelectric materials. For example, the reported ZT values were not highly advantageous.

Third, the authors claimed that the stronger mechanical stiffness is achieved through the cellular design; however, the experimental test on mechanical properties for thermoelectric cellular architectures was too simply conducted, and the stronger mechanical stiffness when compared with other cellular designs was not demonstrated in detail.

Last but not least, the authors hypothesized that the optimum aspect ratio was similar for both hexagonal column and cuboid TE legs; however, their heat spread behaviors are different.

Reviewer #2 (Remarks to the Author):

The authors designed cellular thermoelectric (TE) architectures and synthesized all-inorganic Cu₂Se-based 3D-printing inks for the fabrication of the proposed structures, which achieved superior power output and mechanical property. Moreover, a detailed research was conducted on the rheological properties of the TE inks as well as the preparation technologies. This work not only demonstrates a new architecture that delivers impressive power output, but also provides a feasible strategy to fabricate the cellular honeycomb TE legs. Given the significance and novelty, I recommend this manuscript to be published on Nature Communications journal by referring to the comments as below:

1. Since Se tends to evaporate during sintering, have the authors considered the working stability of Cu₂Se TE legs at such high temperature?
2. Is there a more systematical and widely applicable guidance for the topological design for TE architectures?

Reviewer #3 (Remarks to the Author):

The authors of this paper fabricated thermoelectric devices by 3D printing technology based on Cu₂Se. Several cellular architectures were designed and tested with proper techniques. The thermoelectric

transport properties such as the electrical conductivity, the Seebeck coefficient and the thermal conductivity were measured. The obtained power generation of these device are impressive. I recommend this paper for publication. Here are some comments and suggestions before acceptance.

(1) The Cu₂Se reported in this paper is p-type. In a typical thermoelectric power generator, both p-type and n-type materials are required. Have the authors fabricated the n-type device? If not, please give some comments on n-type Cu₂Se with cellular architectures.

(2) The thermal conductivity were measured by a conventional method which is applicable for isotropic materials. However, the thermal conductivity of hollow hexagonal and honeycomb structures must be anisotropic after sintering. The thermal conductivity in the honeycomb plane should be different from the thermal conductivity perpendicular to it. Please validate the measured results.

(3) What is the role of radiation at high temperature? The radiation effect should be taken into account when measuring thermal conductivity.

Response to the reviewers' comments

The followings are the responses to the reviewers' comments for the manuscript "Cu₂Se-based thermoelectric cellular architectures for efficient and durable power generation."

▪ Reviewer #1

General comment: This manuscript reported the extrusion-based 3D printing fabrication of Cu₂Se-based thermoelectric cellular architectures for power generation application. The cellular design is interesting. However, the manuscript is far from the Nature Communications quality and is more suitable for a specialized journal.

Response: We sincerely appreciate the reviewer's comments on our manuscript as it would consolidate the novel aspects of our study. We truly agree with the reviewer's comments about the necessity for the comparative discussion with the thermoelectric (TE) foams and the measurement of mechanical properties. Nonetheless, **many of the reported TE foams exhibited limited success to have controllability in pore structures**, and furthermore, **the module-level studies have largely underexplored**. Moreover, we significantly revised the manuscript to incorporate the reviewers' comments and included **new data sets and the related discussion** in the revised manuscript, as presented below.

- New tables to list the state-of-the-art TE foams and 3D-printed TE materials and the related discussion to describe the advances of the current study
- New data on the mechanical properties and the related discussion
- New discussion on the cellular architectures
- New discussion on the optimum aspect ratio of cellular TE architectures
- New data on the working stability of the 3D-printed TE materials and the related discussion
- New discussion on potential candidates of n-type pair of Cu₂Se
- New discussion on the homogeneity of the TE properties.

We strongly believe that the revised form of the manuscript has sufficient quality and novelty enough to be published in *Nature Communications*.

Comment 1: First, the background and prior works in the field of cellular thermoelectric materials are poorly reviewed. The authors just simply pointed out the features of cellular designs for some areas without providing specific cons and pros of these designs. To date, some related reports on the thermoelectric foams have been published and what is the difference between the above works and the author's work? The authors should give a more insightful explanation in the manuscript.

Response: We sincerely appreciate the reviewer's comments about the necessity for the comparative discussion on the difference and the technological advances of our designs, compared with the recently reported porous TE materials including TE foams. As the reviewer commented, the TE foams constructed with various classes of materials such as oxides, organics, carbon allotropes, and their hybrids have been recently reported. **Although the reported TE foams can also be categorized into the cellular TE materials in view of their porosities, they are clearly distinct from our cellular honeycomb architecture in terms of i) the purpose and benefit of the structuring, ii) pore structures, iii) fabrication process, and iv) TE properties.** Here, we have listed the recently reported TE foams to compare with our 3D-printed TE cellular architecture in Table A1.

Table A1. Comparison of this work with the recently reported TE foams.

Ref.	Materials	Structuring purpose / benefit	Pore structure	Fabrication method	TE properties (peak ZT)	Remark
[A1]	SiC	Enhancement of TE properties / reduction of thermal conductivity	Stochastic open pores	Macromolecule pyrogenation	ZT: 1.338×10^{-4} S: ~140 σ : ~1.04 κ : ~13	Uncontrollable pore structures
[A2]	SiC+Si	Enhancement of TE properties / NA	Stochastic open pores	Macromolecule pyrogenation	S: ~185 σ : 2.2 PF: 7.8	Uncontrollable pore structures
[A3]	Ca _{0.95} Sm _{0.05} MnO ₃	Using gas heat source / NA	Stochastic open pores	Sacrificial template method	S: ~197	Need template or mold
[A4]	CNT	Flexibility / bendable for 10000 cycles	Stochastic open pores	Rapid solvent evaporation	ZT: 7.6×10^{-4} S: 32.6 σ : 4.02 κ : 0.17	
[A5]	Bi ₂ Te ₃ /cellulose fiber	Flexibility / bendable for 100 cycles	Stochastic open pores	UBMS (Unbalanced magnetron sputtering) on porous cellulose fiber substrate	ZT: 0.38 S: ~134 σ : 2100 κ : 0.46	Need porous substrate
[A6]	PEDOT:PS S	Lightweight with flexibility / NA	Stochastic open pores	Freezing and sublimation of solvent in vacuum (aerogel)	S: ~19 σ : ~220 PF: 6.8	

[A7]	PEDOT:PS S/ melamine foam	Lightweight with flexibility / NA	Stochastic open pores	Dip coating melamine foam into PEDOT:PSS solution	$ZT: 0.007$ $S: 20.3$ $\sigma: 44.3$ $\kappa: 0.0905$	Need porous substrate
[A8]	PANI/CNTs -PDMS	Pressure sensor / dynamic pressure sensing test for 10000 cycle	Stochastic open pores	Salt-leaching method	$S: 17.1$	Need NaCl as a sacrificial template
[A9]	ZnO/graphene	NA	Stochastic open pores	Salt-leaching method	N/A	Need NaCl as a sacrificial template
[A10]	PVDF-PPy- MWCNT	Enhancement of TE properties / reduction of thermal conductivity	Stochastic open pores	Salt-leaching method	$ZT: 1.5 \times 10^{-5}$ $S: 19.8$ $\sigma: 0.055$ $\kappa: 0.044$	Need NaCl as a sacrificial template
[A11]	MWCNT/P VDF & GNP/PVDF	Enhancement of TE properties / reduction of thermal conductivity	Stochastic open pores	Used super critical carbon dioxide (scCO ₂) as a physical blowing agent	$S_{MWCNT/PVDF}: 6-10$ $S_{GNP/PVDF}: 25-58$ $\sigma_{MWCNT/PVDF} : \sim 2 \times 10^{-4}$ $\sigma_{GNP/PVDF} : \sim 3 \times 10^{-7}$ $\kappa_{MWCNT/PVDF} : \sim 0.1$ $\kappa_{GNP/PVDF} : \sim 0.17$	Required high pressure for foam structuring
[A12]	PVDF- MWCNT & PVDF-GNP	Enhancement of TE properties / reduction of thermal conductivity	Stochastic open pores	Salt-leaching method	$ZT: \sim 1 \times 10^{-3}$ $S: \sim 35$ $\sigma: \sim 1$ $\kappa: \sim 0.07$	Need salt as a sacrificial template
[A13]	CNT-Ag nanocomposite aerogels (CNTANAs)	Enhancement of TE properties & Lightweight / reduction of thermal conductivity	Stochastic open pores	Freezing and drying of solvent (aerogel)	$ZT: 0.011$ $S: 54$ $\sigma: \sim 9.5$ $\kappa: \sim 0.075$	Uncontrollable pore structures
This work	Cu ₂ Se	Higher energy conversion efficiency and mechanical durability / higher module power and mechanical stiffness	Periodic closed pores	3D printing	$ZT: 1.21$ $S: 185.4$ $\sigma: 183.42$ $\kappa: 0.52$	Controllable architectures by the 3D printing without any templates or substrates

ZT : dimensionless TE figure of merit, σ : electrical conductivity ($S\ cm^{-1}$), S : Seebeck coefficient ($\mu V\ K^{-1}$), κ : thermal conductivity ($W\ m^{-1}\ K^{-1}$), PF: power factor ($\mu W\ cm^{-1}\ K^{-2}$), T_h : hot-side temperature, T_c : cold-side temperature, E : elastic modulus (MPa), σ_y : yield strength (MPa), σ_f : fracture strength (MPa), CNT: carbon nano tube, PEDOT:PSS: poly(3,4-thienedioxithiophene):poly(4-styrenesulfonate), PANI: polyaniline, PDMS: polydimethylsiloxane, PVDF-PPy-MWCNT: polyvinylidene fluoride-polypyrrole-multi-walled carbon nanotube, GNP: graphene nano-platelets, CNTANAs: CNT-Ag nanocomposite aerogels

We could summarize the technological advances of the current study compared with the reported TE foams as follows.

1. **Most reports of the TE foams generally aim for the enhancement of ZT values in materials. Micro or macroscale stochastic pores in these materials can act as scattering sites of phonons for the reduction of thermal conductivity of materials**, as a result, enhancing their ZT values. So far, numerous studies have been reported for enhancing inherent TE properties of materials by controlling atomic-, nano-, micro-scale structuring. The formation of stochastic pores in the TE foams is another type of strategy for improving the properties of materials by porous structuring. However, the obtained ZT values in these TE foams were not high enough to use them for practical applications. The detailed discussion on the TE properties will be described below. Another benefit that can be obtained from the TE foams of organics- or carbon-based materials is the mechanical flexibility or lightweight in materials, which can find the applications of wearable devices. On the other hand, **in this study, we aim to enhance the energy conversion efficiency and mechanical durability in power generating modules by constructing periodic cellular pore structures**. Our designs realized by the 3D printing process can optimize the system-level thermal transport and structure-induced mechanical stiffness, which clearly distinguish from the previous studies of TE foams with perspectives of the research purpose and resulting benefits.
2. **The previously reported TE foams have stochastic open-cell structures, whereas our work reports well-defined periodic closed-cellular structures**. It is well known that the architecture of porous medium is critical for their mechanical properties including the strength and stiffness^{A14-A19}. In stochastic foams, their mechanical properties exhibit a quadratic dependence on the relative density. If the density is reduced in random foams, their mechanical strengths are rapidly deteriorated^{A14,A16}. However, in closed-cellular structures, the mechanical properties follow a linear scaling relationship with respect to their relative density^{A14,A16}. Therefore, **superior cellular materials can be developed if we exploit the structural advantages of the closed-cellular structure**^{A15,A18,A19}.
3. **Our 3D printing process enables precisely shaping topologies of pores and matrix**, which can realize manufacturing the efficient and durable TE cellular materials having the designed periodic architectures. **This capability of the 3D printing process to construct well-designed porous architectures is impossible to be achieved by traditional processes** such as the use of sacrificial templates or macromolecule pyrogenation that generate only randomly formed stochastic pore structures in materials.
4. We would like to emphasize that **the current study is the first demonstration of the production of cellular TE architecture with a high ZT value**. The peak ZT value of the

3D-printed Cu₂Se in this study marked 1.2, which is at least an order of magnitude higher than the reported values of the TE foams. For example, except 0.38 of Bi₂Te₃/cellulose fiber composite, most of the reported *ZT* values by the TE foams are less than 0.1, of which the material could not guarantee the practical applicability to the power generation and refrigeration.

Accordingly, we included **the Table A1 to further review the field of cellular TE materials** (Supplementary Table 1) in the revised Supplementary Information and **the following related discussion to describe the advances made by our study** in the revised manuscript (page 4-5, 19).

“The shape and layout of pores are critical to the mechanical properties of the cellular materials, since they determine the effective density and deformation modes. For example, in stochastic open-cell structures, structural Young’s modulus decreases with density as an empirical power law with an exponent of two to three. However, in periodic closed-cellular structures such as honeycomb architecture, both the mechanical strength and stiffness linearly decrease with the relative density³⁶⁻³⁸. Thus, the closed-cellular structures tend to exhibit greater mechanical performance than open-cellular structures when their densities are reduced in a similar amount³⁹. Honeycombs represent the two-dimensional (2D) closed-cellular architectures used almost exclusively today due to their relatively simple structures and excellent mechanical properties, such as high in-plane compression and out-of-plane shear properties⁴⁰. When compared with triangular and hexagonal truss-structured materials with similar densities, optimally designed honeycombs exhibited multi-fold enhancement in compressive strength. The honeycomb architecture resisted buckling upon compression unlike the truss-based counterparts⁴¹. Recently, the specific stiffnesses of ceramic hexagonal and triangular honeycombs were reported as $> 10^7$ Pa kg⁻¹ m⁻³ that surpass other micro- and nanoscale lattices of similar relative densities⁴².”

“Recently, the TE foams constructed with various classes of materials such as oxides, organics, carbon allotropes, and their hybrids have been reported. Although the reported TE foams might also be categorized into the cellular TE materials in view of their porosities, they are clearly distinct from our cellular honeycomb architecture in terms of i) the purpose and benefit of the structuring, ii) pore structures, iii) fabrication process, and iv) TE properties (Supplementary Table 3). Especially, most reports of the TE foams generally aim for the enhancement of *ZT* values in materials by the formation of stochastic pores to intensify phonon scattering, which can reduce thermal conductivity of materials. However, in this study, we demonstrated that our designs with periodic pore structures, manufactured by the 3D printing process can optimize the system-level thermal transport and structure-induced mechanical stiffness, improving the efficiency and durability of modules.”

Comment 2: Second, the reported thermoelectric material is common and the performance does not represent a significant advance when compared with state-of-the-art thermoelectric materials. For example, the reported *ZT* values were not highly advantageous.

Response: As the reviewer pointed out, the *ZT* of 3D-printed Cu₂Se is not as high as the state-of-the-

art TE materials such as the recently reported doped or nanostructured Cu chalcogenides. However, we would like to emphasize that **the ZT value of our 3D-printed Cu₂Se is one of the highest among the state-of-the-art 3D-printed TE materials**, as summarized in Table A2. Specifically, we could compare our result with the recently reported Cu_{2-x}S fabricated by the pseudo-3D printing process (ink molding). This material exhibited the highest ZT of 0.63 at around 970 K, about 50% lower than the ZT of 1.2 achieved by our 3D-printed Cu₂Se (Fig. A1).

Table A2 | Comparison of this work with the state-of-the-art 3D-printed TE materials and modules.

Printing method [ref]	Materials	Binder	TE properties	Remark
Pseudo-3D printing [A21]	Cu _{2-x} S	Sodium carboxy methylcellulose (organic)	ZT: 0.63 S: ~249.4 $\mu\text{V/K}$ κ : ~0.36 $\text{W m}^{-1} \text{K}^{-1}$ σ : ~43.3 S cm^{-1}	Molding apparatus was used (not a true 3D printing)
Selective laser meting [A22]	Bi _{0.4} Sb _{1.6} Te ₃	None	ZT: 1.1 S: 95 – 192 $\mu\text{V/K}$ κ : 1.6 $\text{W m}^{-1} \text{K}^{-1}$ σ : 6.05 - 11.8 S cm^{-1}	Stacking of thin layers (cubic shape) Only p type has been produced
Stereo lithography apparatus (SLA) [A23]	Bi _{0.5} Sb _{1.5} Te ₃	Photo-resin (organic)	ZT: 0.12 S: 145 – 190 $\mu\text{V/K}$ κ : 0.2 $\text{W m}^{-1} \text{K}^{-1}$ σ : ~50 S cm^{-1}	Need to remove the photo-resins by thermal decomposition (substrate-assisted bulk shape)
Pseudo-3D printing [A24]	SnSe	Sodium carboxy methylcellulose (organic)	ZT: 1.7 S: 240 $\mu\text{V/K}$ κ : 0.36 $\text{W m}^{-1} \text{K}^{-1}$ σ : 1 – 4 S cm^{-1}	Molding apparatus was used (not a true 3D printing)
Screen printing [A25]	TiS ₂ (n-type)	Hexylamine (organic)	ZT: 0.22 S: 96.6 $\mu\text{V/K}$ κ : 0.69 $\text{W m}^{-1} \text{K}^{-1}$ σ : 544 S cm^{-1}	Thin layer printing of TE ink and construct cuboidal device by folding (p-type module was constructed by conductive polymer, PEDOT)
3D printing [A26]	Bi ₂ Te ₃	Poly(lactide-co-glycolide) (organic)	ZT: 0.08 S: 240 $\mu\text{V/K}$ κ : 0.25 $\text{W m}^{-1} \text{K}^{-1}$ σ : 600 S cm^{-1}	Extrusion of ink into wire shape and device construction
Aerosol jet printing [A27]	Sb ₂ Te ₃	Poly(vinylpyrrolidone) (organic)	ZT: NA S: 83 – 105 $\mu\text{V/K}$ κ : 1 – 6 $\text{W m}^{-1} \text{K}^{-1}$ σ : 300 – 500 S cm^{-1}	Thin layer printing on curved surface

3D printing [A28]	$\text{Bi}_{0.5}\text{Sb}_{1.5}\text{Te}_3$	Poly(vinylpyrrolidone) (organic)	$ZT: 0.104$ $S: 150 \mu\text{V/K}$ $\kappa: 0.543 \text{ W m}^{-1} \text{ K}^{-1}$ $\sigma: 76.2 \text{ S cm}^{-1}$	Claimed as Direct Writing but no 3D construction of device directly by printing
Extrusion printing [A29] 3D	p: $\text{Bi}_{0.4}\text{Sb}_{1.6}\text{Te}_3$ n: $\text{Bi}_2\text{Sb}_{2.7}\text{Se}_{0.3}$	Sb_2Te_4 chalcogenidometallate (inorganic)	$ZT: 0.9 \text{ (p)} / 0.6 \text{ (n)}$ $S: 180\sim 200 \text{ (p)} / -110\sim -140 \text{ (n)} \mu\text{V/K}$ $\kappa: 0.50\sim 0.63 \text{ W m}^{-1} \text{ K}^{-1}$ $\sigma: 550\sim 200 \text{ (p)}, 500\sim 250 \text{ (n)} \text{ S cm}^{-1}$	1 st all-inorganic ink-based 3D printed TE device

ZT : dimensionless TE figure of merit, σ : electrical conductivity (S cm^{-1}), S : Seebeck coefficient ($\mu\text{V K}^{-1}$), κ : thermal conductivity ($\text{W m}^{-1} \text{ K}^{-1}$),

Fig. A1 | ZT comparison of the current work with the reported bulk $\text{Cu}_2\text{Se}^{\text{A20}}$ and pseudo-3D-printed $\text{Cu}_{2-x}\text{S}^{\text{A21}}$.

Compared with the reported TE foams, the ZT value of our 3D-printed Cu_2Se is at least an order of magnitude higher than those, as summarized in Table A1. This extraordinary high performance could be attributed to the material's inherent properties of Cu_2Se , considering the reported cellular porous materials are generally based on oxides, organics, and carbon allotropes not having high TE properties intrinsically. This limitation observed in the previous studies originates from the underlying challenges in creating controlled pores in inorganic TE materials by the conventional procedures. In this study, we addressed this challenge by the development of fully inorganic inks and the 3D printing process for fully inorganic Cu_2Se TE materials. Thereby, we firstly achieved not only the fabrication of cellular periodic architecture but also the high ZT value of Cu_2Se TE materials, which will accelerate the expansion of the field of TE cellular architectures.

Further, we could compare our result with the TE properties of the pure Cu₂Se bulk materials fabricated by the traditional melting method^{A20}. As seen in Fig. A1, **the ZT of the 3D-printed Cu₂Se shows the identical temperature-dependent behaviours to those of the reported Cu₂Se bulk**, indicating the high quality of materials. Moreover, although the ZT value of the 3D-printed Cu₂Se in this study is ~20% lower in the entire temperature range, this reduction of the properties can be understood with the consideration of the fabrication processes of the ink-3D printing and subsequent pressureless sintering. Moreover, combining the benefit of cellular architectures that is impossible to be achieved by the traditional processes (26% increase of the energy conversion efficiency), **the lowered ZT value in the 3D-printed Cu₂Se can be compensated by 3D-printed architecturing**.

Accordingly, we included Table A1 and A2 for the comparison of TE properties in the revised Supplementary Information and the following discussion in the revised manuscript (page 16).

“This maximum value is one of the highest among the reported TE materials prepared from various inks or by 3D printing process (Supplementary Table 2). Compared with the recently reported TE foams, the ZT value of our 3D-printed Cu₂Se is at least an order of magnitude higher than those, as summarized in Supplementary Table 3.”

Comment 3: Third, the authors claimed that the stronger mechanical stiffness is achieved through the cellular design; however, the experimental test on mechanical properties for thermoelectric cellular architectures was too simply conducted, and the stronger mechanical stiffness when compared with other cellular designs was not demonstrated in detail.

Response: We agree with the reviewer that the experimental measurement and comparative discussion of mechanical properties would be crucial to demonstrate the concept of our study. Accordingly, we conducted the compression test on the 3D-printed honeycomb Cu₂Se to evaluate its mechanical properties and compared it with that of the cuboid. As shown in Fig. A2, **the stress-strain curve of the 3D-printed honeycomb in the elastic region exhibit similar behaviour to that of the cuboid (Fig. A2). The calculated modulus 939 MPa, which was almost identical to 911 MPa of the cuboid. Interestingly, the honeycomb exhibited larger plastic deformation region and higher fracture strain by three times, compared with the cuboid.** This improvement may originate from unique structural characteristics of the honeycomb, which can distribute the stress concentration into a whole structure^{A30-A32}. Sun et al. demonstrated the distribution of the stress concentration in honeycomb

architectures by the finite element model (FEM) simulations and experiments (Fig. A2)^{A32}.

Fig. A2 | Stress-strain curve of 3D-printed Cu_2Se cuboid and honeycomb calculated with the material's cross-sectional area by compression stress test.

[Redacted]

Fig. A3 | Numerical simulation of crushing processes of honeycomb structure by the FEM^{A32}.

We further estimated the modulus of the honeycomb with the effective cross-sectional area including pores. The calculated modulus was 174.7 MPa, almost identical to the predicted value of 157.2 MPa by the Gibson-Ashby relation, showing the structural quality of the 3D-printed architectures. Such superior mechanical properties of honeycomb closed-cellular architecture against other cellular designs have been studied in the previous works^{A14-A19}. First, the mechanical properties of cellular structures have been rigorously studied. The empirical power law between the density and mechanical properties (e.g.,

structural Young's modulus, strength and stiffness) derived by Gibson and Ashby has been widely validated and used^{A14-A19}. According to the Ashby-Gibson relations, in stochastic open-cell structures, structural Young's modulus decreases with density as an empirical power law with an exponent of two to three. However, in periodic closed-cellular structures such as honeycomb architecture, both the mechanical strength and stiffness linearly decrease with the relative density^{A14-A16}. Thus, **the closed-cellular structures tend to exhibit greater mechanical performance than open-cellular structures when their densities are reduced in a similar amount**^{A17}. Particularly, **when compared with triangular and hexagonal truss-structured materials with similar densities, optimally designed honeycombs exhibited multi-fold enhancement in compressive strength**. The honeycomb architecture resisted buckling upon compression unlike the truss-based counterparts^{A18}. Recently, the specific stiffnesses of ceramic hexagonal and triangular honeycombs were reported as $> 10^7$ [Pa/(kg/m³)] that surpass other micro- and nanoscale lattices of similar relative densities^{A19}.

[Redacted]

Fig. A4 | Ashby-Gibson plot for Young's modulus as a function of density^{A14}. Stochastic foams follow an empirical power law with an exponent of two to three. However, closed, hierarchical porous materials such as woods follow an empirical power law with an exponent of one.

Accordingly, we included **the previous works and the discussion which justifies the selection of honeycomb closed-cellular structures in the revised introduction**, as follows. Also, we **included the stress-strain curve of the 3D-printed honeycomb** (Fig. A2) in the revised Supplementary Information

(Supplementary Fig. 3) and **the related following discussion** in the revised manuscript (page 8).

“The mechanical properties of the 3D-printed Cu₂Se were measured by compressive test under the uniaxial compression mode on a cuboid and a honeycomb. The stress-strain curve of the 3D-printed honeycomb in the elastic region exhibit similar behaviour to that of the cuboid (Supplementary Fig. 3). The calculated modulus 939 MPa, which was almost identical to 911 MPa of the cuboid. Interestingly, the honeycomb exhibited larger plastic deformation region and higher fracture strain by three times, compared with the cuboid. This improvement may originate from unique structural characteristics of the honeycomb, which can distribute the stress concentration into a whole structure⁵²⁻⁵⁴. Sun et al. demonstrated the distribution of the stress concentration in honeycomb architectures by the finite element model (FEM) simulations and experiments⁵⁴. Based on the measured properties, we predicted the compressive stiffness of the Cu₂Se-honeycomb architecture using the Ashby-Gibson relation (Fig. 1h). As the number of unit cell in the honeycomb architecture increases, effective compressive stiffness decreases in the range of 100 – 200 MPa due to the decrease in density (ρ), agreeing with the measured value of 174.7 MPa in the honeycomb with the effective cross-sectional area including pores.”

Comment 4: Last but not least, the authors hypothesized that the optimum aspect ratio was similar for both hexagonal column and cuboid TE legs; however, their heat spread behaviors are different.

Response: We agree that the optimum aspect ratio would be different for hexagonal column and cuboid TE legs if the exact physics occurring at TE leg-electrode interface is considered. First, the **thermal contact resistance** depends on the heat spread behaviour (shown in Fig. 1D and 1E). Particularly in a hexagonal module, the heat spread behaviour is affected by the wall thickness and the area enclosed by the hexagonal column. However, in our design stage, **it was difficult to account for the effect of heat spread behaviour, because it was unknown how the thermal contact resistance would change for various hexagonal cross-sectional shapes**. Second, the **electrical contact resistance** also depends on the cellular TE leg’s geometry because the solder exhibits different properties after curing for various TE leg’s wall thicknesses. In actual experiments, **the electrical contact resistance of the hexagonal column device was measured to be smaller than that of the cuboid TE leg**. The cuboid TE leg with the largest thickness presented the greatest electrical contact resistance.

During the design stage, we scoped the objective of this work as demonstrating the 3D-printed TE cellular modules that were optimized by available knowledge and theories. Unfortunately, **the knowledge of the heat spread behaviours and solder properties as a function of the module cross-sectional shapes and areas was unavailable, but it was discovered as a result of this research**. As our next step, we will accurately optimize the aspect ratio of the cellular TE legs by considering the contact physics. Through our future investigation, we will offer empirical correlations between the cellular TE-leg geometries and contact resistances.

We include **the following discussion on the optimum aspect ratio** in the revised manuscript (page 7-8).

“To accurately optimize the aspect ratio, the electrical and thermal contact resistances at TE leg-electrode interface need to be considered. When electrical current or heat transfers from the TE leg to an electrode, the cross-sectional shape of the TE leg affects how the current or heat spreads into the electrode⁵¹. Furthermore, solder properties may be influenced by the TE leg thickness, if the TE leg is not sufficiently thick. If empirical correlations between the TE leg geometry and contact resistances are available for the cellular or other architectures, the topology of the 3D TE module will be more accurately determined. For the honeycomb architecture, we select a relatively small aspect ratio, because it is challenging to achieve the optimum $//A$ in actual device, as fabricating high-aspect-ratio, multiple hexagonal columns is not feasible yet.”

References

- [A1] Wei W. et al. Macrostructural influence on the thermoelectric properties of SiC ceramics. *Scr. Mater.* **57**, 1081-1084 (2007).
- [A2] Wei W. et al. The influence of Si distribution and content on the thermoelectric properties of SiC foam ceramics. *Microporous Mesoporous Mater.* **112**, 521-525 (2008).
- [A3] Reddy E. S. et al. Open porous foam oxide thermoelectric elements for hot gases and liquid environments. *Energy Convers. Manag.* **48**, 1251-1254 (2007)
- [A4] Lee M.-H. et al. Freely shapable and 3D porous carbon nanotube foam using rapid solvent evaporation method for flexible thermoelectric power generators. *Adv. Energy Mater.* **9**, 1900914 (2019).
- [A5] Jin Q. et al. Cellulose fiber-based hierarchical porous bismuth telluride for high-performance flexible and tailorable thermoelectrics. *ACS Appl. Mater. Interfaces* **10**, 1743-1751 (2018).
- [A6] Gordon M. P. et al. Soft PEDOT:PSS aerogel architectures for thermoelectric applications. *J. Appl. Polym. Sci.* **134**, 44070 (2017).
- [A7] Thongkham W. et al. Conductive nanofilm/melamine foam hybrid thermoelectric as a thermal insulator generating electricity: theoretical analysis and development. *J. Mater. Sci.* **54**, 8187-8201 (2019).
- [A8] Wang Y. et al. 3D geometrically structured PANI/CNT-decorated polydimethylsiloxane active pressure and temperature dual-parameter sensors for man-machine interaction applications. *J. Mater. Chem. A.* **8**, 15167-15176 (2020).
- [A9] Zhao H. et al. Conjoined photo-thermoelectric effect in ZnO-graphene nanocomposite foam for self-powered simultaneous temperature and light sensing. *Sci. Rep.* **10**, 11864 (2020).
- [A10] Aghelineja M. et al. Fabrication of open-cell thermoelectric polymer nanocomposites by template-assisted multi-walled carbon nanotubes coating. *Compos. Part B* **145**, 100-107 (2018).
- [A11] Sun Y.-C. et al. Study on the thermoelectric properties of PVDF/MWCNT and PVDF/GNP composite foam. *Smart Mater. Struct.* **24**, 085034 (2015).
- [A12] Aghelinejad M. et al. Thermoelectric nanocomposite foams using non-conducting polymers with hybrid 1D and 2D nanofillers. *Materials* **11**, 1757 (2018).

- [A13] Sun X. et al. Thermoelectric performance of conducting aerogels based on carbon nanotube/silver nanocomposites with ultralow thermal conductivity. *RSC Adv.* **6**, 109878-109884 (2016).
- [A14] Gibson L.J. and Ashby M.F., *Cellular Solids: Structure and Properties*. (Cambridge University Press, 2001)
- [A15] Schaedler T.A. et al. Ultralight metallic microlattices. *Science* **334**, 962-965 (2011).
- [A16] Yeo S. J., Oh M. J. and Yoo P. J. Structurally controlled cellular architectures for high performance ultra lightweight materials. *Adv. Mater.* **31**, 1803670 (2019).
- [A17] Berger J. B., Wadley H. N. G. and McMeeking R. M. Mechanical metamaterials at the theoretical limit of isotropic elastic stiffness. *Nature*, **543**, 533–537 (2017).
- [A18] Bauer J. et al. High-strength cellular ceramic composites with 3D microarchitecture. *Proc. Natl. Acad. Sci. U.S.A.*, **111**, 2453-2458 (2014).
- [A19] Muth J. T. et al. Architected cellular ceramics with tailored stiffness via direct foam writing. *Proc. Natl. Acad. Sci. U.S.A.* **114**, 1832-1837 (2017).
- [A20] H. Liu et al. Copper ion liquid-like thermoelectrics. *Nature mater.* **11**, 422-425 (2012).
- [A21] Burton M. R. et al. Earth abundant, non-toxic, 3D printed Cu_{2-x}S with high thermoelectric figure of merit. *J. Mater. Chem. A*, **7**, 25586-25592 (2019).
- [A22] Qiu, J. et al. 3D Printing of highly textured bulk thermoelectric materials: mechanically robust BiSbTe alloys with superior performance. *Energy & Environ. Sci.* **12**, 3106-3117 (2019).
- [A23] He, M. et al. 3D printing fabrication of amorphous thermoelectric materials with ultralow thermal conductivity. *Small* **11**, 5889-5894 (2015).
- [A24] Burton, M. et al. 3D printed SnSe thermoelectric generators with high figure of merit. *Adv. Energy Mater.* **9**, 26 (2019).
- [A25] Rösch, A. et al. Fully printed origami thermoelectric generators for energy-harvesting. *npj Flex. Electron.* **5**, 1-8 (2021).
- [A26] Peng, J. et al. 3D extruded composite thermoelectric threads for flexible energy harvesting. *Nat. Commun.* **10**, 5590 (2019).
- [A27] Dun, C. et al. 3D printing of solution-processable 2D nanoplates and 1D nanorods for flexible thermoelectrics with ultrahigh power factor at low-medium temperatures. *Adv. Sci.* **6**, 1901788 (2019).

- [A28] Su, N. et al. 3D-printing of shape-controllable thermoelectric devices with enhanced output performance. *Energy* **195**, 116892 (2020).
- [A29] Kim, F. et al. 3D printing of shape-conformable thermoelectric materials using all-inorganic Bi₂Te₃-based inks. *Nat. Energy* **3**, 301-309 (2018).
- [A30] Meo M. et al. Numerical simulations of low-velocity impact on an aircraft sandwich panel. *Compos. Struct.* **62**, 3-4 (2003).
- [A31] Lu C. et al. Stress distribution on composite honeycomb sandwich structure suffered from bending load. *Procedia Eng.* **99**, 405-412 (2015).
- [A32] Sun G. et al. Experimental and numerical study on honeycomb sandwich panels under bending and in-panel compression. *Mater. Des.* **133**, 154-168 (2017).

▪ Reviewer #2

General comment: The authors designed cellular thermoelectric (TE) architectures and synthesized all-inorganic Cu₂Se-based 3D-printing inks for the fabrication of the proposed structures, which achieved superior power output and mechanical property. Moreover, a detailed research was conducted on the rheological properties of the TE inks as well as the preparation technologies. This work not only demonstrates a new architecture that delivers impressive power output, but also provides a feasible strategy to fabricate the cellular honeycomb TE legs. Given the significance and novelty, I recommend this manuscript to be published on Nature Communications journal by referring to the comments as below:

Comment 1: Since Se tends to evaporate during sintering, have the authors considered the working stability of Cu₂Se TE legs at such high temperature?

Response: We appreciate the reviewer's fruitful comment. To demonstrate the working stability of the 3D-printed Cu₂Se material, we characterized the electrical conductivity and Seebeck coefficient through multiple heat cycles in the temperature range from room temperature to 873 K, which was the highest hot-side temperature of the module for the power generation measurement. Fig. B1 shows the thermoelectric (TE) properties of the 3D-printed Cu₂Se during the heat cycle by three times. **In the cycle test, the electrical conductivities and Seebeck coefficients were well preserved without degradation within the equipment error range** (SBA458 Nemesis, Netzsch, Germany), demonstrating the working stability of our samples.

Fig. B1 | Heat cycle performance of the 3D-printed Cu_2Se samples. Temperature-dependent (a) electrical conductivity, and (b) Seebeck coefficient.

We further tested the working stability of our samples for a longer duration time by measuring the TE properties of the samples during the thermal annealing at 873 K. As shown in Fig. B2, **the 3D-printed Cu_2Se sample maintained both the primary electrical conductivity and the Seebeck coefficient for 1 h without thermal degradation.** These results clearly demonstrated the working stability of our 3D-printed Cu_2Se for the power generation application.

Fig. B2 | Working stability of the 3D-printed Cu_2Se material. a, Electrical conductivity and b, Seebeck coefficient of the 3D-printed samples at room temperature during the heat treatment at 873 K for 1 h.

We included the heat cycle performance and working stability data in the revised Supplementary Information (Supplementary Fig. 15 and 16) and the following discussion in the revised manuscript (page17-18).

“To further demonstrate the working stability of the 3D-printed Cu₂Se material, we characterized the electrical conductivity and Seebeck coefficient through multiple heat cycles in the temperature range from room temperature to 873 K, which was the highest hot-side temperature of the module for the power generation measurement. In the cycle test, the electrical conductivities and Seebeck coefficients were well preserved without degradation within the equipment error range by three times (Supplementary Fig. 15). Moreover, we tested the working stability of the 3D-printed samples for a longer duration time by measuring their TE properties during the thermal annealing at 873 K. As shown in Supplementary Fig. 16, the 3D-printed Cu₂Se sample maintained both the primary electrical conductivity and the Seebeck coefficient for 1 h without thermal degradation. These results clearly demonstrated the working stability of our 3D-printed Cu₂Se for the power generation application.”

Comment 2: Is there a more systematical and widely applicable guidance for the topological design for TE architectures?

Response: We appreciate the valuable suggestion by the reviewer. **To design the cellular TE architecture in a systematic and widely applicable manner, it is desired to develop a correlation between the cellular TE-leg geometries and contact resistances.** Theoretical models have been reported for predicting the TE power outputs of TE legs, although the equation-based models are mostly applicable to only simple geometries. However, to accurately design the cellular-TE model design, the physics occurring at TE leg-electrode interface must be considered as we discuss in the manuscript. First, the thermal contact resistance depends on the heat spread behaviour (shown in Fig. 1D and 1E in the revised manuscript). Particularly in a hexagonal module, the heat spread behaviour is affected by the wall thickness and the area enclosed by the hexagonal column. Second, the electrical contact resistance also depends on the cellular TE leg’s geometry because the solder exhibits different properties after curing for various TE leg’s wall thicknesses. If the correlations that predict the thermal and electrical contact resistances as a function of the module cross-sectional shapes and areas are developed, we will be able to integrate these correlations with the existing TE models for TE modules. As our next step, we plan to develop the correlations for the contact resistances and use the models to systematically explore the design space for cellular TE legs. Fig. B3 briefly illustrates our tentative approach for designing the cellular TE architectures.

Fig. B3 | Tentative approach for designing the cellular TE architectures.

To design the topological design for the TE architecture in a large design space, topology optimization may be considered. The topology optimization identifies non-intuitive and mathematically optimal structures, and has been employed widely for mechanical structural systems. However, the conventional topology optimization requires an extensive computation resources, and often results in complex designs that cannot be fabricated by legacy techniques. Thus, the topology optimization is not yet considered as widely applicable approach in the field.

Accordingly, **we included the following discussion on the topological designs of TE architectures** in the revised manuscript (page 19).

“To further optimize the topological design for the TE architecture in a large design space, topology optimization may be considered. The topology optimization identifies non-intuitive and mathematically optimal structures, and has been employed widely for mechanical structural systems. However, the conventional topology optimization requires an extensive computation resources, and often results in complex designs that cannot be fabricated by legacy techniques. Thus, the topology optimization is not yet considered as widely applicable approach in the field.”

▪ Reviewer #3

General comment: The authors of this paper fabricated thermoelectric devices by 3D printing technology based on Cu₂Se. Several cellular architectures were designed and tested with proper techniques. The thermoelectric transport properties such as the electrical conductivity, the Seebeck coefficient and the thermal conductivity were measured. The obtained power generation of these device are impressive. I recommend this paper for publication. Here are some comments and suggestions before acceptance.

Comment 1: The Cu₂Se reported in this paper is p-type. In a typical thermoelectric power generator, both p-type and n-type materials are required. Have the authors fabricated the n-type device? If not, please give some comments on n-type Cu₂Se with cellular architectures.

Response: We appreciate the reviewer's fruitful comment. **The family of copper chalcogenides is known to intrinsically exhibit the properties of p-type semiconductors** because the point defect of Cu vacancy that acts as an electron acceptor is easily formed by the composition control or oxidation. Accordingly, to the best of our knowledge, **there are no studies to report n-type electrical transport of Cu₂Se compounds**. Recently, a few studies have been reported on n-type semiconductors of ternary or quaternary copper chalcogenide compounds of CuAgSe, CuFeS₂, and Cu_{1-x}Zn_xFeS₂^{C1,C2}. However, the *ZT* values of these materials are less than 0.6 at the highest, not high enough to use them as an n-type pair of our Cu₂Se. **As alternatives, different classes of n-type TE semiconductors such as doped SnSe, skutterudites, half-Heusler can be candidates as an n-type pair of our 3D-printed Cu₂Se^{C3}.**

Accordingly, we included the following sentence to describe the potential candidates of n-type pairs for Cu₂Se in the revised manuscript (page 16).

“Recently, a few studies have been reported on n-type semiconductors of ternary or quaternary copper chalcogenide compounds of CuAgSe⁶⁷, CuFeS₂, and Cu_{1-x}Zn_xFeS₂⁶⁸. However, the *ZT* values of these materials are less than 0.6 at the highest, not high enough to use them as an n-type pair of our Cu₂Se. As alternatives, different classes of n-type TE semiconductors such as doped SnSe, skutterudites, half-Heusler can be potential candidates as an n-type cellular pair of our 3D-printed Cu₂Se.”

Comment 2: The thermal conductivity were measured by a conventional method which is applicable for isotropic materials. However, the thermal conductivity of hollow hexagonal and honeycomb structures must be anisotropic after sintering. The thermal conductivity in the honeycomb plane should be different from the thermal conductivity perpendicular to it. Please validate the measured results.

Response: Thermal conductivity is the inherent properties of materials, not dependent on the geometries of materials. However, we agree that **the anisotropy in the thermal conductivity can be induced by two possibilities of crystallographic orientations and homogeneity in TE properties of the samples.** Various workhorse TE materials of Bi₂Te₃, BiSbTe, and SnSe have strong anisotropies in crystal structures and the related TE properties including thermal conductivity. For example, the hot-pressed Bi₂Te₃ usually exhibits anisotropic TE properties along the pressing directions due to the crystallographic textures. However, **Cu₂Se has an isotropic crystal structure of the face-centred cubic above 100~150 °C, where the phase transition occurs from the monoclinic to the cubic phase.** Hu et al. demonstrated that all TE properties of the hot-pressed Cu₂Se material in the previous report are not dependent on the pressing directions, **indicating the isotropic TE properties independent of the crystallographic orientation**^{C4}. The other possibility to result in the anisotropy in TE properties can be the homogeneity in the 3D-printed samples, depending on dimensions and shapes. However, **we already validated the homogeneity of TE properties in the various 3D-printed samples** in Fig. 4b in the manuscript, which allow us to rule out this possibility. In fact, compared with anisotropic TE materials such as Bi₂Te₃, BiSbTe, and SnSe, such an isotropy in TE properties of Cu₂Se TE material is recognized as an advantage in processing (e.g. dicing) for manufacturing TE modules because the arrangement of anisotropic TE legs in a module require additional processing cost.

Accordingly, **we included the following sentence to emphasize the isotropy of TE properties of Cu₂Se** in the revised manuscript (page 14).

“Although Cu₂Se is known to have the isotropic TE properties owing to the isotropic crystal structure of the face-centred cubic above 100~150 °C, the inhomogeneity of the TE properties can be caused by the inhomogeneity of the distribution of temperatures and particle density in the dried specimen during the sintering.”

Comment 3: What is the role of radiation at high temperature? The radiation effect should be taken into account when measuring thermal conductivity.

Response: We appreciate the thoughtful comment by the reviewer. Based on equation (1), we evaluated the thermal conductivity of the 3D-printed Cu₂Se material by measuring bulk density (ρ), the thermal diffusivity (D), and the specific heat capacity (C_p).

$$\kappa = \rho C_p D \quad (1)$$

In these three parameters, **the radiation effect on the thermal conductivity is already included in the thermal diffusivity, which was measured by the laser flash analysis (LFA)**. The LFA method is well established and widely utilized to measure the thermal properties of materials within the accuracy of ~3% in a wide range of temperatures (room temperature~1250 °C in our equipment (LFA 467 HT, Netzsch) and thermal conductivities (0.1 W m⁻¹ K⁻¹ to 2000 W m⁻¹ K⁻¹). The LFA is conducted by detecting rising temperatures at the rear face during the heat absorption at the front face by a pulse of light. The radiational heat loss of the sample in the LFA method is expressed by Newtonian cooling law and the Biot number. **Applying the following Biot numbers to the algorithms, the heat diffusion is calculated from the data obtained by the LFA method taking into radiant heat loss**^{C5-C10}. The Biot number (Y_x for front face and Y_r for rear face) is defined as equation (2) and (3),

$$Y_x = 4\varepsilon\sigma T_0^3 d / \lambda \quad (2)$$

$$Y_r = 4\varepsilon\sigma T_0^3 r_0 / \lambda \quad (3)$$

where ε is the total hemispherical emissivity of the specimen, σ is the Stefan–Boltzmann constant, T_0 is the steady state temperature of the specimen before pulse heating, d is the thickness of the specimen, r_0 is the radius of the specimen, λ is the thermal conductivity of the specimen and, respectively. As a result, an analysis algorithm using the curve-fitting method matches the all-region of the temperature hysteresis curve as a theoretical solution under the heat loss boundary condition, finally evaluating thermal diffusivity. **Accordingly, the equipment error range by the radiation heat loss is within 0.35% at maximum, negligible in the evaluation of thermal conductivity**^{C11}.

We included the following sentence to describe the radiation loss in the revised manuscript (page 22).

“The equipment error range of thermal diffusivity by the radiation heat loss is within 0.35% at maximum⁷².”

References

- [C1] Wang X. et al. Compound defects and thermoelectric properties in ternary CuAgSe-based materials. *J. Mater. Chem. A* **3**, 13662-13670 (2015).
- [C2] Xie H. et al. The Role of Zn in Chalcopyrite CuFeS₂: Enhanced Thermoelectric Properties of Cu_{1-x}Zn_xFeS₂ with In Situ Nanoprecipitates. *Adv. Energy mater.* **112**, 521-525 (2008).
- [C3] Qiu P. et al. High-efficiency and stable thermoelectric module based on liquid-like materials. *Joule* **3**, 1538–1548 (2019).
- [C4] Hu T. et al. One-step ultra-rapid fabrication and thermoelectric properties of Cu₂Se bulk thermoelectric material. *RSC Adv.* **9**, 10508-10519 (2019).
- [C5] Watt D. A. Theory of thermal diffusivity by pulse technique, *Br. J. Appl. Phys.* **17**, 231 (1966).
- [C6] Heckman R. C. Finite pulse-time and heat-loss effects in pulse thermal diffusivity measurements. *J. Appl. Phys.* **44**, 1455-1460 (1973).
- [C7] Cowan R. D. Pulse method of measuring thermal diffusivity at high temperatures. *J. Appl. Phys.* **34**, 926-927 (1963).
- [C8] Cape J. A. and Lehman G. W. Temperature and finite pulse-time effects in the flash method for measuring thermal diffusivity. *J. Appl. Phys.* **34**, 1909-1913 (1963).
- [C9] Clark L. M. III and Taylor R. E. Radiation loss in the flash method for thermal diffusivity. *J. Appl. Phys.* **46**, 714-719 (1975).
- [C10] Baba T. and Ono A. Improvement of the laser flash method to reduce uncertainty in thermal diffusivity measurements. *Meas. Sci. Technol.* **12**, 2046-2058 (2001).
- [C11] Nishi T., Azuma N. and Ohta H. Effect of radiative heat loss on thermal diffusivity evaluated using normalized logarithmic method in laser flash technique. *High Temp. Mater. Process* **39**, 390-394 (2020).

REVIEWERS' COMMENTS

Reviewer #1 (Remarks to the Author):

I would like to recommend acceptance.

Reviewer #2 (Remarks to the Author):

The authors have satisfactorily responded the comments and I am glad to recommend the publication.

Reviewer #3 (Remarks to the Author):

All my comments have been addressed. This paper should be published as it is.

Response to the reviewers' comments

The followings are the responses to the reviewers' comments for the manuscript "Cu₂Se-based thermoelectric cellular architectures for efficient and durable power generation."

▪ Reviewer #1

Comment: I would like to recommend acceptance.

Response: We sincerely appreciate the reviewer's positive assessment of our manuscript.

▪ Reviewer #2

Comment: The authors have satisfactorily responded the comments and I am glad to recommend the publication.

Response: We sincerely appreciate the reviewer's positive assessment of our manuscript.

▪ Reviewer #3

Comment: All my comments have been addressed. This paper should be published as it is.

Response: We sincerely appreciate the reviewer's positive assessment of our manuscript.